# Estimating Local AgGDP across the World

Yating Ru[1,*], Brian Blankespoor[2,*], Ulrike Wood-Sichra[3], Timothy S. Thomas[3], Liangzhi You[3], and Erwin Kalvelagen[3]

[1]Cornell University
[2]World Bank
[3]International Food Policy Research Institute
[*]These authors contributed equally to this work.

**Correspondence:** Brian Blankespoor (bblankespoor@worldbank.org)

**Abstract.** Economic statistics are frequently produced at an administrative level such as the sub-national division. However, these measures may lack sufficient local variation for effective analysis of local economic development patterns and exposure to natural hazards. Agriculture GDP is a critical indicator for measurement of the primary sector, on which more than 2.5 billion people depend on their livelihoods, and it provides a key source of income for the entire household (FAO, 2021). Through a data fusion method based on cross-entropy optimization, this paper disaggregates national and sub-national administrative statistics of agricultural GDP into a global gridded dataset at approximately 10 x 10 kilometers for the year 2010 using satellite-derived indicators of the components that make up agricultural GDP, namely crop, livestock, fishery, hunting and forestry production. To illustrate the use of the new dataset, the paper estimates the exposure of areas with at least one extreme drought during 2000 to 2009 to agricultural GDP, which amounts to around US$432 billions of agricultural GDP circa 2010, with nearly 1.2 billion people living in those areas. The data are available on the World Bank Development Data Hub (DOI: http://doi.org/10.57966/0j71-8d56; IFPRI and World Bank, 2022).

## 1 Introduction

According to the Food and Agriculture Organization of the United Nations, at least 2.5 billion people depend on the agricultural sector for their livelihood and it provides a key source of employment and income for the poor and vulnerable people (FAO, 2013, 2019, 2021). Yet, economic statistics of the agricultural sector are frequently produced at a national or lower administrative level and may not adequately capture local variation in production activities. Furthermore, a geographic unit of interest, such as the natural area of a river basin, may not align with political administrative boundaries, limiting the ability to conduct a comprehensive overlay analysis of the area. Lastly, local conditions can pose challenges to measurement across the world. Around five billion hectares of land is dedicated to agriculture, but collecting and reporting data across the world can be challenging, especially in areas affected by fragility, conflict, and violence, which can result in incomplete or outdated geographic coverage.

Detailed agricultural data are critical to examining a wide range of agricultural issues including technology and land use (e.g. Bella and Irwin, 2002; Luijten, 2003; Staal et al., 2002; Samberg et al., 2016), exposure to natural hazards (e.g. Murthy

et al., 2015), evaluation of forest restoration opportunities (Shyamsundar et al., 2022) as part of nature-based climate solutions (Griscom et al., 2017), and patterns of and productivity of economic development (e.g. Nelson, 2002; Elhorst and Strijker, 2003; Gollin et al., 2014; Reddy and Dutta, 2018). Carrão et al. (2016) examine the exposure of people and economic activity to drought using measures of physical elements (e.g. cropland and livestock). Rentschler and Salhab (2020) find that low and middle-income countries have 89% of global flood exposed population and poor people account for almost 600 million, who are directly exposed to the risk of intense flooding. Vesco et al. (2021) examine linkages between climate variability and agricultural production as well as conflict. They find that climate variability contributes to an increase in the spatial concentration of agricultural production within countries. Furthermore, in countries with a high share of agricultural employment in the national workforce, they find this combined effect increases the likelihood of conflict onset. To better target rural development strategies for economic growth and poverty reduction, as well as conserve the natural resource base for long-term sustainable development, we need to accurately delineate the spatial distribution of agricultural resources and production activities (Wood et al., 1999).

One method to address the case where administrative boundaries and geographic areas of interest are not aligned is to use the gridded (raster) data format. It provides an intermediate and consistent unit for disaggregation and aggregation (e.g. UNISDR, 2011). Data-disaggregation methods can use detailed data to inform estimates of aggregated data from large areas at the local level (e.g. see review in Pratesi et al., 2015). Several spatial data products from global models are available to estimate population at a local level (see review in Leyk et al., 2019).

Previous evidence-based risk analyses take advantage of global data of hazards to estimate exposure of population and economic activity (e.g. Gunasekera et al., 2015, 2018; Ward et al., 2020; Rentschler and Salhab, 2020). Gross Domestic Product (GDP) is a critical economic indicator in the measurement and monitoring of an economy in a country that is typically only available at national and occasionally sub-national levels. Regional indicators play a key role in the necessary variation to forecast regional GDP (Lehmann and Wohlrabe, 2015) and food security (Andree et al., 2020). Previous efforts to estimate local GDP use high resolution spatial auxiliary information such as luminosity or population data to provide local variation. Methods by Nordhaus (2006); World Bank and UNEP (2011); Kummu et al. (2018); Murakami and Yamagata (2019) took advantage of gridded population data, which is the result of a model disaggregating the most detailed level population data into grids. However, income is not evenly distributed among people nor infrastructure (Berg et al., 2018). In fact, the divide between the rich and poor is even widening in our time (Dabla-Norris et al., 2015). The method used in World Bank and UNEP (2011) stratifies the population by rural and urban, yet the definition of these geographic areas can vary based on the selection of the population model (Leyk et al., 2019). These measurements matter in application to stylized facts such as the strong negative correlation of the level of urbanization with the size of its agricultural sector (Roberts et al., 2017). Also, the strong assumption of uniform distribution of labor in agriculture is another key concern (Gollin et al., 2014). Uneven agricultural productivity across different regions or locations can lead to a non-uniform distribution of labor within the sector, which has implications for the accuracy and effectiveness of models based on rural per capita allocation. Other methods used land cover such as vegetation and built-up indices, however did not incorporate types of agriculture like cropland and livestock (Gunasekera et al., 2015; Goldblatt et al., 2019).

Other methods to estimate GDP at a local level take advantage of nighttime lights datasets. Doll et al. (2006) and Elvidge et al. (2009) found nighttime lights to provide a uniform, consistent, and independent estimate for economic activity, and several other studies (e.g. Chen and Nordhaus, 2011; Henderson et al., 2012; Ghosh et al., 2010; Bundervoet et al., 2015; Wang et al., 2019; Eberenz et al., 2020; Wang and Sun, 2021) utilized this striking correlation between luminosity and economic activities to estimate economic output on the ground. While night light is a good reflection of economic activities in manufacturing and urban areas, night light data may not capture the agricultural activity as it requires areas to emit light. Bundervoet et al. (2015) suggest that agricultural indicators rather than rural population could improve the estimation of GDP given the importance of agriculture in many of the economies in their sample of Africa. Gibson et al. (2021) find that night time lights data are a poor predictor of economic activity in low population density rural areas.

In this paper, we present a high resolution gridded Agricultural GDP (corresponding to the "agriculture, forestry, and fishing, value added" in World Development Indicators, henceforth AgGDP) dataset that is produced through a spatial allocation model by distributing national and sub-national statistics to 5-arcminute grids based on satellite-derived information of constituents of AgGDP, including forestry, hunting, and fishing, as well as cultivation of crops and livestock production[1]. Our main contribution is to construct a global dataset of gridded AgGDP. This entails a massive effort of data collection and integration. We extend and apply the cross-entropy framework developed in the Spatial Production Allocation Model (SPAM) for crops that pioneered the use of cross-entropy optimization in spatial allocation (You and Wood, 2003; You et al., 2014, 2018; Yu et al., 2020). We construct and integrate global datasets of the components of AgGDP as priors and then reconcile the values with the regional account statistics using cross-entropy optimization. As an illustration of the novel dataset, we assess the exposure of economic activity to natural hazards with a focus on AgGDP. Significant progress has been made to measure physical assets such as built-up area along with its importance in population models (Rubinyi et al., 2021) and estimate hazards in order to quantify the exposure to natural hazards (e.g Gunasekera et al., 2015; UNDRR, 2019). However, the detailed spatial distribution of AgGDP is less known. So, we apply these data to inform efforts quantifying the population and AgGDP at risk to drought and water scarcity highlighting a linkage to a subset of agricultural activities as well as an association with population.

The rest of this paper is structured as follows. The next section provides a detailed description of the methodology and data. Then, we present the model results, uncertainty, and validation. Afterwards, we demonstrate one possible application by analyzing AgGDP exposure to natural hazards. Finally, we provide concluding remarks.

## 2 Methodology and data

Following the composite structure of AgGDP, we disaggregate the national and sub-national statistics into a global grid through a cross-entropy allocation model. Given the limited availability of data and the global scope of the study, we made various efforts to adjust official statistics and create priors for different components based on the available data. Below we discuss the construction of each component, AgGDP statistics and the allocation model followed by the global natural hazards data. Given

---

[1]Agriculture, forestry, and fishing corresponds to ISIC divisions 1-3 and includes forestry, hunting, and fishing, as well as cultivation of crops and livestock production

the spatial resolution and year of reference of the input data for the crop value of production, we estimate AgGDP for the year 2010 into 5-arcminute grids (10x10 km) across the world.

## 2.1 Construction of components

For each pixel, we construct an estimated value of production based on high spatial resolution information of the five components that serve as priors in the modeling process: crop, livestock, forestry, fishing, and hunting. Given the lack of information on the hunting component, we disaggregate the forestry component into two parts: timber and non-timber products of forestry. The non-timber products of forestry includes an even distribution of hunting. The construction of the five components is described below in four subsections: crop, livestock, forestry (timber and non-timber) and fishing.

### 2.1.1 Crop value of production

The prior for the crop component in the gridded AgGDP is generated by multiplying the quantity of production from the global SPAM 2010 version 1 dataset[2] (You et al., 2018) with producer prices at the country level from FAOSTAT (FAO, 2016) for each crop and then summed together. As for the producer prices, ideally, we need sub-national level figures since prices for agricultural products can vary greatly within countries and their subdivisions, but such a dataset is not available globally. Therefore, we use the FAOSTAT's national producer prices and take the average of 2009-2011, in order to mitigate the potential impact of temporal variation. However, due to missing data for certain countries, crops, and years, this average may be based on a smaller time period or the closest year available. As mentioned earlier, SPAM is a cross-entropy model, which calculates a plausible allocation of crop areas and production to approximately 10 km pixels, based on agricultural statistics at national and sub-national levels, combined with gridded layers of cropland, irrigated areas, population density and potential crop areas and yields (Yu et al., 2020). SPAM's output distinguishes between 42 crops (33 individual crops, 9 aggregated crops) that together add up to practically all cultivated crops in a country with four parameters including production, yield, physical area and harvest area.

For aggregated SPAM crops (such as other cereals, other pulses, vegetables, fruits, etc.), we computed their prices by taking the weighted average of their components, as follows:

$$Price_{Jagg} = \frac{\Sigma_j price_j prod_j}{\Sigma_j prod_j}, \forall j \in Jagg \tag{1}$$

where $Jagg$ is the aggregated crop group, $j$ is any crop that belongs to $Jagg$, $Price_{Jagg}$ is the price of the aggregated crop group, $price_j$ is the price of crop $j$, and $prod_j$ is the production of $j$.

For each grid, the value of crop production is thus:

$$Cropval_i = \Sigma_j prod_{i,j} price_j, \forall j \text{ that grow in pixel } i \tag{2}$$

where $Cropval_i$ is the value of total crop production in pixel $i$, $prod_{i,j}$ is the production of crop $j$ in pixel $i$, and $price_j$ is the price of crop $j$. A map of global gridded crop production value as a prior is shown in Figure 1.

---

[2]Available at www.mapSPAM.info

**Figure 1.** The assembled crop production value used as a prior in the cross-entropy model. Sources: FAO (2016); Yu et al. (2020); Authors' calculation (2022)

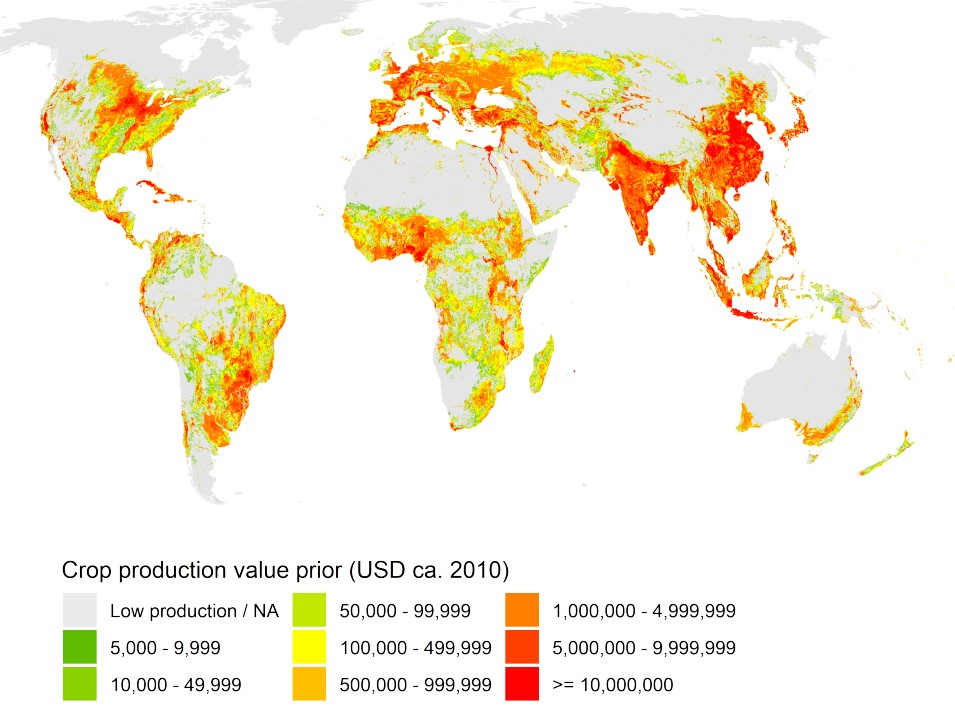

Crop production value prior (USD ca. 2010)

| | | |
|---|---|---|
| Low production / NA | 50,000 - 99,999 | 1,000,000 - 4,999,999 |
| 5,000 - 9,999 | 100,000 - 499,999 | 5,000,000 - 9,999,999 |
| 10,000 - 49,999 | 500,000 - 999,999 | >= 10,000,000 |

### 2.1.2 Livestock production

Livestock accounts for an estimated 40% of the global value of agriculture output and plays an important role in ensuring the livelihoods and food security for over one-sixth of the world's population (FAO, 2018). Yet, it is still under rapid expansion as the global demand for animal-sourced products such as meat, milk, eggs, and hides continues to grow (Herrero and Thornton, 2013). While species and quantities of livestock raised vary among regions and husbandry farmers, there are five primary species - cattle, sheep, goats, pigs, and chicken - that prevail worldwide and provide essential products for human consumption.

We calculate the prior for the component of livestock production in gridded AgGDP based on the distribution maps of the above five primary species from the Gridded Livestock of the World (Robinson et al., 2014; Gilbert et al., 2018) and FAOSTAT's value of production of livestock products (including meat, milk, eggs, honey and wool) (FAO, 2020). Due to data limitations, distribution maps for other animals such as ducks, horses, camels, and bees are not available. But the FAOSTAT's livestock production values include a more comprehensive list of animals and their products. By distributing FAOSTAT values to grids in proportion to the five primary livestock species, we assume that other animals included in FAOSTAT have a similar spatial distribution to the five primary livestock species. This assumption is generally valid, but may not be accurate in special areas, such as deserts where camels are an important source of livestock products. To facilitate comparison, the animal-specific

density numbers are converted to one animal type by using International Livestock Units as conversion factors (Eurostat, 2018), as shown in Table 1. The conversion factors reflect biomass differences between different animals [3]. Then the densities of the animal equivalent values are multiplied by the total area of each 5-arcminute pixel to get the count of animals per grid, which is used to calculate the share of animal counts and then multiplied by the FAOSTAT's value of production to obtain the livestock production prior for each pixel.

$$lsval_i = lsval_x \frac{lsnum_i}{\Sigma_X lsnum_i}, \forall i \in X \tag{3}$$

where $lsval_i$ is the total value of livestock production in pixel i; $lsval_x$ is the value of livestock production (meat, milk, eggs, honey and wool) that is reported at the national level; $lsnum_i$ is the total number of equivalent animals in pixel $i$; and $X$ is a set including all pixels that fall within the boundary of a nation.

A map of global gridded livestock production value as a prior is shown in Figure 2.

**Table 1.** Conversion factors for different livestock types. Sources: Eurostat (2018)

| Livestock type | Conversion factor |
| --- | --- |
| Cattle | 1 |
| Pigs | 0.3 |
| Goats | 0.1 |
| Sheep | 0.1 |
| Chicken | 0.01 |

### 2.1.3 Forestry production and hunting

People have utilized forest resources for a long time throughout history for their livelihood and various other purposes (Hossain et al., 2008). Up until now, over a billion people still rely on forest resources for food security and income generation to some extent (FAO, 2018). In the world's least developed regions, 34 countries depend on fuelwood to provide more than 70% of energy, among which 13 nations require 90% of energy (FAO, 2018).

The contribution of forest production to AgGDP can be classified into two broad types: wood (logging) products and non-wood forest products. Wood (logging) products are the most exploited commodities in the forestry sector. The trees are harvested for fuelwood and industrial roundwood, which is processed into a variety of products including lumber, plywood, furniture, and paper products. Non-wood forest products are defined by the Food and Agriculture Organization of the United

---

[3]The uniform conversion factors may oversimplify local variation in livestock patterns. Future work may consider using country-specific values of livestock products from FAOSTAT.

**Figure 2.** The assembled livestock production value used as a prior in the cross-entropy model. Sources: Robinson et al. (2014); Gilbert et al. (2018); Eurostat (2018); Authors' calculation (2022)

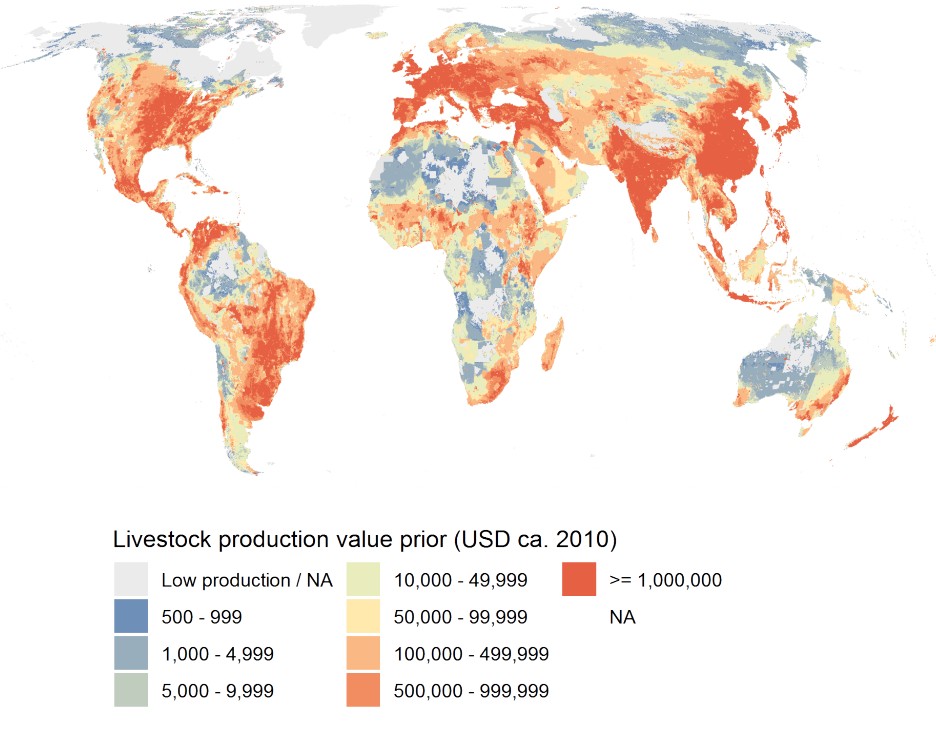

Livestock production value prior (USD ca. 2010)

| | | |
|---|---|---|
| Low production / NA | 10,000 - 49,999 | >= 1,000,000 |
| 500 - 999 | 50,000 - 99,999 | NA |
| 1,000 - 4,999 | 100,000 - 499,999 | |
| 5,000 - 9,999 | 500,000 - 999,999 | |

Nations (FAO).[4] It is estimated that millions of households around the world depend on non-wood forest products for their livelihood. Some 80% of people in the developing world use these products in their everyday life (Sorrenti, 2016).

For a complete assessment of forest production priors, this study takes both wood and non-wood products into consideration. The gridded non-wood forest products dataset used in this study was jointly developed by Resources for the Future and the World Bank (Siikamäki et al., 2015) through an approach of meta-regression modeling, which integrates over 100 estimates at various locations from a literature review and multifold information on ecological and socioeconomic factors. The value of non-wood forest products is resampled to the 5-arcminute grid cell size and converted to 2010 USD for consistency with other

AgGDP components. As part of non-timber products, we include hunting with an even distribution across units and time given the lack of information.

The value of wood products prior per pixel is calculated based on forest loss from year 2010 to year 2011 excluding loss due to fire, with an assumption that the forests were mainly cut down for timber production. The Moderate Resolution Imaging Spectroradiometer (MODIS) Land Cover map (Friedl et al., 2010) for year 2011 is overlaid on top of that for year 2010 to

---

[4]These products are "goods of biological origin other than wood derived from forests, other wooded land and trees outside forests", including foods (nuts, fruits, mushrooms, etc.), food additives (herbs, spices, sweeteners, etc.), fibers (for construction, furniture, clothing, etc.), and plant and animal products with chemical, medical, cosmetic or cultural value.

detect the area that has changed from forest to non-forest.[5] However, forest loss due to fire needs to be removed because it does not result in timber production in most cases[6]. Thus, fire information for year 2010 is obtained from the NASA Fire Information for Resource Management System (FIRMS) (NASA, 2018) and areas that experienced forest fires are eliminated. After the identification of the forest area change in each pixel, the value of wood production at the national level is taken from a FAO led project (Lebedys and Li, 2014) and proportionally disaggregated to arrive at a pixel-wise value of wood products as follows:

$$Woodval_i = (forestval_x - nonwoodval_x)\frac{forestloss_i}{\Sigma_X forestloss_i}, \forall i \in X \tag{4}$$

where $Woodval_i$ is the value of wood products in pixel i; $forestval_x$ is the value of forest products reported at the national level; $nonwoodval_x$ is the value of non-wood products at the national level which is derived from Siikamäki et al. (2015); $forestloss_i$ is the area of forest loss excluding loss to fire in pixel $i$; again, $X$ is a set including all pixels that fall within the boundary of a nation.

In our analysis of the forestry sector GDP, we have utilized the estimates provided by Lebedys and Li (2014) as the best available source. However, it should be noted that these estimates primarily capture activities within the formal forestry sector and do not take into account the value-added generated by informal activities such as wood fuel production and non-wood forest products. To account for non-timber forest products, we have utilized the estimates provided by Siikamäki et al. (2015). Despite these efforts, it is acknowledged that the current analysis may still underestimate the forestry sector GDP due to the lack of reliable data on fuel wood production, which could account for half of global wood harvests (Ghazoul and Evans, 2004). This is a common issue as fuel wood values are often not properly captured in official statistics, as they are often collected for subsistence or sold in remote rural areas in many countries (Lebedys and Li, 2014). In future research, we intend to make efforts to acquire more reliable data on fuel wood production to improve the accuracy of our estimates of the forestry sector GDP.

A map of global gridded wood forest production value as a prior is shown in Figure 3.

### 2.1.4  Fishery production

Fish makes up approximately 17% of animal-sourced protein in the human diet worldwide (Mathiesen, À. M., 2018). The fishery industry supports the livelihood of 12% of world population by creating 200 million jobs along its value chain. In the global trade system, 80 billion USD worth of fish is exported from developing countries and it plays a crucial role in promoting local economic development (Kelleher et al., 2009).

We estimate both freshwater inland fisheries and marine production values using the FISHSTAT (FAO, 2009) data with a classification based on the fish production categories. The inland fishery production value is the result of disaggregating corresponding country level statistics in proportion to areas of inland water bodies in the 5-arcminute pixel. This is a simplified

---

[5]The measurement is limited to detection of land cover change from satellites and might not fully account for selective harvesting or forest degradation. And the area of forest is considered homogeneous of equal production value. Also, it could result in upward bias when trees are cut down for plantation replanting and not used in further processing of timber production.

[6]Still, sometimes wood harvests may occur after forest fires, and therefore the elimination could underestimate the area harvested for wood products.

**Figure 3.** The assembled wood forest production value used as a prior in the cross-entropy model. Sources: Friedl et al. (2010); Siikamäki et al. (2015); NASA (2018); Authors' calculation (2022)

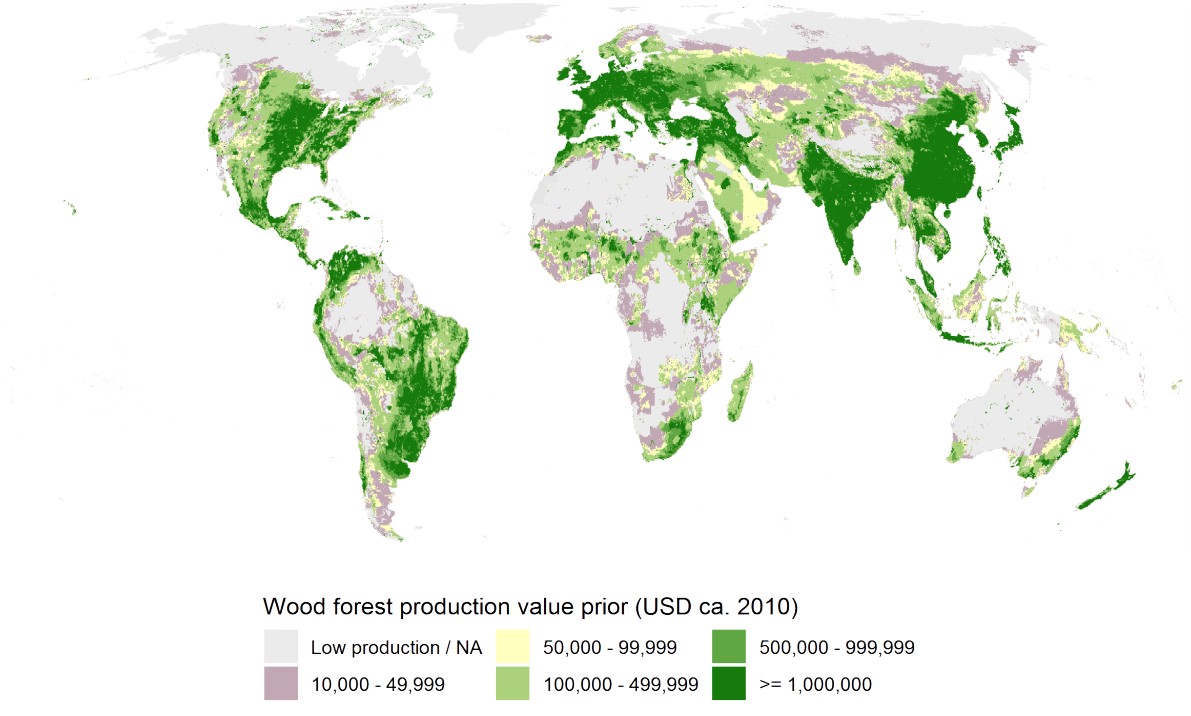

Wood forest production value prior (USD ca. 2010)

| | | |
|---|---|---|
| Low production / NA | 50,000 - 99,999 | 500,000 - 999,999 |
| 10,000 - 49,999 | 100,000 - 499,999 | >= 1,000,000 |

assumption and may cause overestimation in places where there are inland waterbodies, but not much fishery activities going on. The distribution of inland water bodies is obtained from the ESA-CCI (Lamarche et al., 2017). Thus, the value of inland fishing production in each grid is calculated as follows:

$$fishval_i = freshval_x \frac{waterbody_i}{\Sigma_X waterbody_i}, \forall i \in X \tag{5}$$

where $fishval_i$ is the value of fishery production in pixel $i$; $freshval_x$ is the value of fresh fish production at the national level which is aggregated from FISHSTAT; $waterbody_i$ is the area of water bodies in pixel $i$; and $X$ is a set including all pixels $i$ that fall within the boundary of a nation $x$.

The value of marine fisheries production is determined by its proximity to fish landing ports and a composite indicator that equally weighs the number of vessel visits and the total holding capacity of the fishing vessels. We use the port database from the World Port Index (National Geospatial-Intelligence Agency, 2019) and the number of port visits with a vessel hold of fishing vessels from Hosch et al. (2019) to create a composite variable as the prior based on the sum (for each port) of the number of visits (each event in the database) and total vessel hold at the port. The geographic coverage of the ports is calculated for each port using the minimum port distance provided in Hosch et al. (2019). Any distances greater than 150 km were considered as 150 km in this analysis. The value of marine fishing production in each grid is calculated as follows:

$$marineval_i = marineval_x \frac{portindex_i}{\sum_X portindex_i}, \forall i \in X \tag{6}$$

where $marineval_i$ is the value of fishery production in pixel $i$; $portindex_i$ is an equally weighted composite index of the number of visits and the total vessel hold in pixel $i$; and $X$ is a set including all pixels $i$ that fall within the boundary of a nation $x$.

A map of global gridded fishery production value as a prior is shown in Figure 4.

**Figure 4.** The assembled fishery production value used as a prior in the cross-entropy model. Sources: FAO (2009); Lamarche et al. (2017); Hosch et al. (2019); National Geospatial-Intelligence Agency (2019); Authors' calculation (2022)

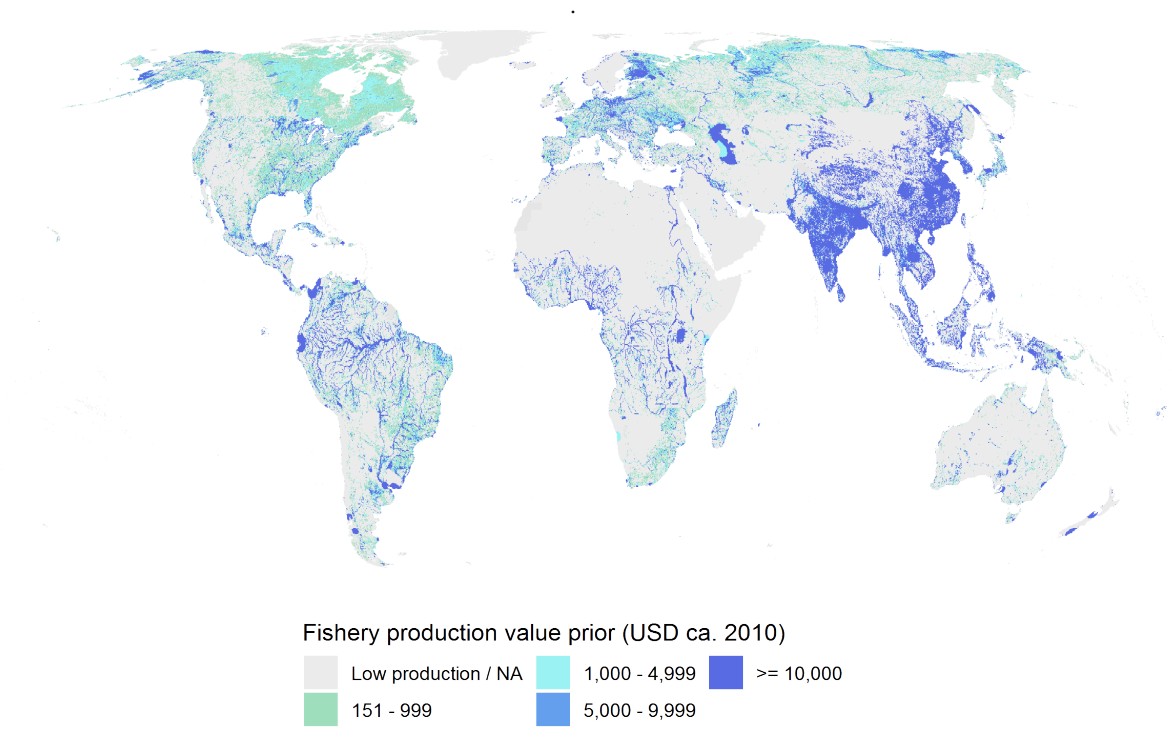

## 2.2    AgGDP Statistics and Linked Grids

Substantial efforts have been made to collect and organize national and sub-national statistics from a variety of sources, including national ministries and reports. However, not every country publishes its AgGDP figures at the sub-national (regional) level and there exist different methods of regionalization including top-down, bottom-up and mixed methods (Eurostat, 2013).[7]

---

[7]Regional Gross Domestic Product (RGDP) can be estimated following the production, income or expenditure approaches. However, RGDP is not typically compiled using the expenditure approach due to the scarcity of data such as inter-regional purchases and sales, or regional exports/imports. On the production

Our database has 68 countries that have sub-national AgGDP data, expressed in varying domestic currencies and for different years. The typical administrative level is at the state or provincial level. Table B7 lists these countries and descriptive statistics including the temporal coverage and the number of sub-national regions at an administrative geographic level including the NUTS level.[8]

To overcome discrepancies in temporal coverage and currency terms (constant and current), and to keep the data consistent and comparable for countries across the world, shares from sub-national statistics are calculated and then applied to a national total to derive a calibrated number at the sub-national level. The national totals are obtained from the publicly available World Development Indicators (WDI) (World Bank, 2019) and averaged over three years around 2010. For a few countries, which do not report their national AgGDP in the WDI database, sums of all AgGDP components are used as proxies.

The World Bank compiles these national accounts data following the International Standard Industrial Classification (ISIC) divisions 1-3 that includes agriculture, forestry and fishing. Given the challenges of compiling national accounts data across the world, limitations include the exclusion of unreported economic activity in the informal or secondary economy. In particular, agricultural output in developing countries may not be reported due to issues such as, natural losses, self-consumption or not exchanged for money. Despite best efforts, agricultural production may be estimated indirectly leading to approximations that are different than the true values. [9]

The calibrated statistics are then linked to grids through a shapefile of the Global Administrative Unit Layers (GAUL) that maintains global geographic layers with a consistent and comprehensively unified coding system (FAO, 2015). Then, we overlay the GAUL administrative boundaries on the grid network to assign the corresponding codes of the administrative units to each grid. For areas where sub-national AgGDPs have different administrative areas than GAUL, the GAUL areas are merged or split to match the sub-national AgGDP areas.

## 2.3   Spatial Allocation Model

After constructing all the components, we define a spatial allocation model in a cross-entropy framework following (You et al., 2014) to allocate administrative statistics to 5-arcminute pixels[10]. National and sub-national AgGDP values are used as a constraint, while the distribution of crop, livestock, fishery, and forestry production (hunting is included in non-timber products of forestry) is used to create priors for estimating pixel-level AgGDP. In actuality, the priors that we have constructed do not encompass all elements of AgGDP, and the national and sub-national AgGDP statistics include a broader range of production values. But the priors account for most variation between pixels, and thus their shares can serve as appropriate proxies in the AgGDP disaggregation model. Lastly, measurement units are unified using deflators and exchange rates.[11]

---

and income approaches, the estimate of market activities is typically from the production approach, whereas the estimate of non-market industries is from the income approach.

[8]The European Union developed a standard for administrative levels: The Classification of Territorial Units for Statistics (NUTS, for the French nomenclature d'unités territoriales statistiques).

[9]See World Bank WDI for more details on metadata and limitations

[10]A comprehensive presentation of the cross-entropy method is in Rubinstein and Kroese (2004)

[11]The currency varies by source. Crops are in local currency. Livestock are in International USD 2004-2006. Fish are USD 2009. Non-timber forest products are in USD 2012 and Timber (forest) are in USD 2011.

The first step is to transform all real-value parameters into corresponding probabilities. Let $S_i$ be the share of the total AgGDP allocated to pixel $i$ within a country $x$. $AgGDP_{i,x}$ is the AgGDP allocated to pixel $i$ in country $x$ and X is a set including all pixels that fall within the boundary of a nation. Therefore:

$$S_i = \frac{AgGDP_{i,x}}{\Sigma_X AgGDP_{i,x}}, \forall i \in X \tag{7}$$

Let *PreAgGDP_i* be the pre-prior allocation of AgGDP share from our best estimate. The first approximation can be done by summing all five calculated pixel level components of AgGDP:

$$PreAgGDP_i = Crop_i + Livestock_i + Forestry_i + Fishing_i + Hunting_i \tag{8}$$

where we assume hunting occurs in areas with equal probability.

     Theoretically, the sum of these components should be close to the official values obtained from the World Development

Indicators. However, it should be noted that due to limitations in available data, we have some components in output values (crop, livestock, and fishery) whereas others in value added (forestry and hunting). This may result in discrepancies and inconsistencies. Overall, we make sure that the official AgGDP values are guaranteed to be no less than the sum of all five components of AgGDP.

$$AgGDP_x = \Sigma_{i \in x} PreAgGDP_i \tag{9}$$

Then, we rescale the prior *AgGDP* to be consistent with the official AgGDP value:

$$PriorAgGDP_i = \frac{PreAgGDP_i AgGDP_x}{\Sigma_i PreAgGDP_x} \tag{10}$$

Then we calculate the prior for S$_i$ as a probability by normalizing PriorAgGDP:

$$PreAlloc_i = \frac{PriorAgGDP_{i,x}}{\Sigma_{i \in X} PriorAgGDP_i} \tag{11}$$

Finally, we formulate a cross entropy model in the following mathematical optimization framework:

$$MIN\ CE(S_i) = \Sigma_i S_i log(S_i) - \Sigma_i S_i log(PreAlloc_i) \tag{12}$$

Subject to the following three conditions:

$$\Sigma_i S_i = 1 \tag{13}$$

$$\Sigma_{i \in k}(\Sigma AgGDP)S_i = SubAgGDP_k\ \forall k \tag{14}$$

$$0 \leq S_i \leq 1\ \forall i \tag{15}$$

where $i$: i=1,2,3,... are pixel identifiers within the allocation unit (e.g. Brazil); and $k$: k=1,2,3, ... are identifiers for sub-national geopolitical units (e.g. a state) where AgGDP values ($SubAgGDP_k$) are available. The objective function is defined as the cross entropy of AgGDP shares and their priors. The first constraint (Equation 13) is the pycnophylactic or volume-preserving constraint (e.g. Tobler, 1979) that ensures the sum of all allocated AgGDP values is equal to the total AgGDP of the country. The next equation (14) sets the sum of all allocated AgGDP within those sub-national units with available data to be equal to the corresponding sub-national AgGDP values. The last equation (15) is a natural constraint for the share of AgGDP to be between 0 and 1, which is also the probability in the cross-entropy model. The modeling framework is flexible in that more constraints can be added if more data are available and/or more reasonable assumptions on how AgGDP should be spatially disaggregated are discovered.[12] Last but not least, we multiply the total regional AgGDP by the probability in the cross-entropy model to derive the final pixel level AgGDP:

$$AgGDP_i = \Sigma_i AgGDP_x S_i \tag{16}$$

## 3  Results, Uncertainty, and Validation

### 3.1  Results

Figure 5 illustrates the result of the cross-entropy model in a global map of gridded AgGDP. The global gridded AgGDP for the year 2010 in 2010 US dollars is in gridded (raster) format at a resolution of 5 arcminute, which approximates to 10 km.[13] The spatial extent and quantity distribution of AgGDP over the world are in agreement with general knowledge of agricultural technology adoption and suitability, with well-known agricultural nations, such as India, China and the United States standing out as regions with relatively high AgGDP compared with many other areas of the world. A number of European countries also exhibit high AgGDP values, which is likely due to the benefit of adopting mechanized farming and technological facilitation, considering that the shares of agricultural land and agrarian population are relatively low in these well-developed places. Countries in Sub-Saharan Africa remain low in agricultural production, as indicated by low-value pixels sparsely spreading over the continent. Within the continent, agricultural production activities primarily take place in geographic areas with suitability and access to markets (e.g. land cultivation see Berg et al., 2018).

We examine the correlation of the AgGDP dataset with two commonly used global datasets to proxy economic activity: night time lights and population. Night time lights data are commonly used in the estimation of local human development and economic activity (e.g. Ghosh et al., 2010; Henderson et al., 2012; Bundervoet et al., 2015; Kummu et al., 2018; Bruederle and Hodler, 2018). We use the sum of the radiance calibrated data for 2010 from the F16 satellite to quantify the correlation

---

[12]For instance, market access may play a role in determining the spatial distribution or spatial structure of AgGDP and can be included as a constraint in the model. However, we provide a parsimonious model without market access.

[13]The coordinate system is the standard WGS84 and saved in GeoTIFF format. For presentation in the paper, the coordinate system of the maps is Eckert IV and transformed from the geographic coordinates in R software. The data are publicly and freely available through the World Bank Development Data Hub website at http://www.doi.org/10.57966/0j71-8d56.

**Figure 5.** Global gridded AgGDP circa 2010 from the Cross-Entropy model in 2010 USD. Source: Authors' calculation (2022)

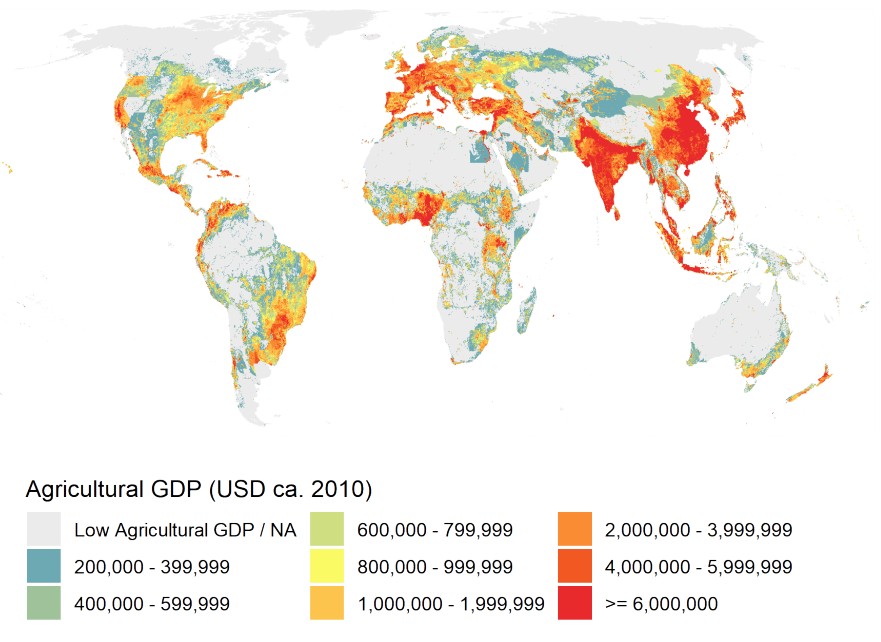

Agricultural GDP (USD ca. 2010)

| | | |
|---|---|---|
| Low Agricultural GDP / NA | 600,000 - 799,999 | 2,000,000 - 3,999,999 |
| 200,000 - 399,999 | 800,000 - 999,999 | 4,000,000 - 5,999,999 |
| 400,000 - 599,999 | 1,000,000 - 1,999,999 | >= 6,000,000 |

between AgGDP and nighttime lights by geographic regions of the world defined by the World Bank.[14] We use rural population derived from Center for International Earth Science Information Network - CIESIN - Columbia University (2017) following methods in Thomas et al. (2019). We use country level data from the World Bank World Development Indicators (World Bank, 2019). We find that the correlation of AgGDP with night light varies across world regions, with Sub-Saharan Africa and the Other Region showing lower correlation values (Table 2). Most World Bank regions have similar patterns of correlation with nighttime lights across the measures of AgGDP and population. Likewise, World Bank income groups show similar patterns across the measures with lower middle and upper middle income groups having higher correlations than low and high income groups. However, notable differences of the correlations exist between geographic levels. The mean correlation of AgGDP with night time lights (NTL) and population (pop) derived from administrative level 2 data is lower than the national level, which presents evidence of new information from the AgGDP dataset.

Furthermore, limitations exist with these commonly used datasets for applications of AgGDP. For night time lights, Li et al. (2020) provide a cautionary note about rural applications where the presence of agricultural activities typically takes place. A population model assumes proportional activity to population by strata (e.g. rural), which does not account for the type of rural of agricultural activity, and the model requires a standard definition of rural, which can pose challenges in global applications (e.g. stylized facts in the urban and development economics literature Roberts et al., 2017). Notably, the rural population

---

[14]Specifically, we use the version 4 product from the F16 satellite (20100111 - 20101209) available at: https://ngdc.noaa.gov/eog/dmsp/download_radcal.html

| AgGDP correlations by | (1) | (2) | (3) | (4) |
| World Bank Regions | NTL (adm 0) | NTL (adm 2) | POP (adm 0) | POP (adm 2) |
| --- | --- | --- | --- | --- |
| AFR | 0.682 | 0.314 | 0.934 | 0.673 |
| EAP | 0.956 | 0.493 | 0.979 | 0.739 |
| ECA | 0.818 | 0.546 | 0.914 | 0.611 |
| LAC | 0.949 | 0.605 | 0.947 | 0.720 |
| MENA | 0.798 | 0.556 | 0.953 | 0.638 |
| Other | 0.896 | 0.669 | 0.909 | 0.697 |
| SOA | 0.929 | 0.547 | 0.929 | 0.716 |

**Table 2.** Spearman correlation of AgGDP with night time lights at the Admin 0 level (1) and Admin 2 level (2) as well as rural population at the Admin 0 level (3) and the Admin 2 level (4), grouped by World Bank Region where AFR is Sub Saharan Africa; EAP is East Asia and Pacific; ECA is Eastern Europe and Central Asia; LAC is Latin America; MENA is Middle East and North Africa; SOA is South Asia and Other is the category for the remaining countries. Sources: NOAA (2011); World Bank (2019); Authors' calculation (2022).

dataset also has variation in the geographic level of the input information, which informs the estimates of population model, and currency across the world, especially when dependent on the frequency of production and availability of a population census. Also, the AgGDP dataset may attenuate modeling concerns of endogeneity when using AgGDP along with population or night time lights.

### 3.2 Fitness-for-use and uncertainty

We provide descriptive statistics of the data and modeling from a fitness-for-use perspective (e.g. Leyk et al., 2019). The data are most appropriate for applications at global, continental and regional scales (You and Wood, 2006). However, decisions regarding the use of the data at smaller spatial extents should be made with caution and with consideration of the underlying assumptions and characteristics of the area in question. Users should take into account factors such as area of the grid cell of AgGDP, the number of subdivisions of AgGDP from the political area (e.g. country), and assumptions in the priors (e.g. see shares of priors in Table B8). When input data contains multiple observations, the AgGDP dataset may still be suitable for use, as it is already standardized in grid cells, which may facilitate integration with other data. As the spatial refinement of ancillary data advances along with greater currency, coverage and representativeness, we expect validation possibilities to increase and inform a better understanding of the uncertainty and the associated fitness-for-use. Also, we intend to improve spatial and temporal coverage when it is feasible.

The process of disaggregating the data from the source level to the target level does impose spatial relationships and is prone to error (Li et al., 2007) and the Modifiable Areal Unit Problem (MAUP) (Openshaw, 1981). In previous work, our team conducted sensitivity analyses and examined consequences of methodological-data choices involved in a cross-entropy model to disaggregate crop production statistics (Joglekar et al., 2019). These analyses included eight scenarios that varied in allocation methods, data grouping, input variables, and different levels of statistics. The analysis indicated that allocation results

are most dependent on the degree of disaggregation and quality of the underlying national and sub-national production statistics.
Therefore, we provide more discussion in section 3.2.1 Regional accounts. Additionally, the results are moderately sensitive to allocation methods. We previously compared three models for the case of Brazil (Thomas et al., 2019) and found that cross-entropy is the most appropriate method for the global study with relatively high accuracy and flexible data requirements, when compared with either the spatial regression or rural population methods. Interested readers may find more details in the Brazil paper. Lastly, the results are somewhat sensitive to the grouping and format of input components that serve as priors, which we discuss in 3.2.2 Components.

### 3.2.1 Regional accounts

The measurement of GDP is challenging (Angrist et al., 2021), especially agricultural production (Carletto et al., 2015). The level of uncertainty associated with these results includes the thematic, spatial and temporal accuracies. We collected regional accounts by sector from various sources into a global database. The data are not balanced over time nor at the geographic level. The variation in the reference year of the regional accounts data influences the temporal balance of the database. This mismatch can influence the regional distribution of the AgGDP that may be different than the target reference year of 2010. Given climate[15] and specifically rainfall is important input to crop and livestock production and may contribute to variation across years (Stanimirova et al., 2019; Zhang et al., 2020), we attempt to reduce this source of error by averaging over multiple years when data are available, which is a similar approach to You et al. (2014). However, this does not eliminate this mismatch. The availability of data varies when grouped by World Bank income (low or lower middle, upper middle and high income). The average absolute temporal difference (ATD) defined as the mean difference in years between the reference regional accounts and the target year (2010) is higher in the low and lower middle income group. Likewise, the mean deviation of the share of AgGDP by country over the year(s) is larger in low or lower middle compared to high income.

The global regional account database includes national and sub-national units at various administrative levels.[16] Following Robinson et al. (2014) in their assessment of Gridded Livestock Of the World (GLW) 2.0, we summarize the average spatial resolution (ASR) of the input regional data, which is the square root of the land area divided by the number of administrative units (See Figure 6). We find that on average the ASR value increases from high to low income groups based on World Bank 2010 classifications. Following Yu et al. (2020) we suggest that users can view the ASR map as an indicator of uncertainty level since the model is proven most dependent on the ASR of statistics. Larger ASR represents more sparsity of input statistics and more uncertainty of the gridded results.

### 3.2.2 Components

Another source of uncertainty is the indirect temporal inaccuracy propagated from the input datasets of the components, which are modeled. We discuss all five components of AgGDP: crop, livestock, forest, fish and hunting. The SPAM model (You et al., 2014) is a result of several gridded modeled datasets including rural population density from Global Rural-Urban

---

[15]For a discussion on climate yield factors see Block et al. (2008).

[16]This also includes cases where administrative units at the same level are merged to match the geography of the regional accounts data.

**Figure 6.** The average spatial resolution of the regional accounts data by country. Sources: World Bank (2019); Authors' calculation (2022) and various sources see Appendix.

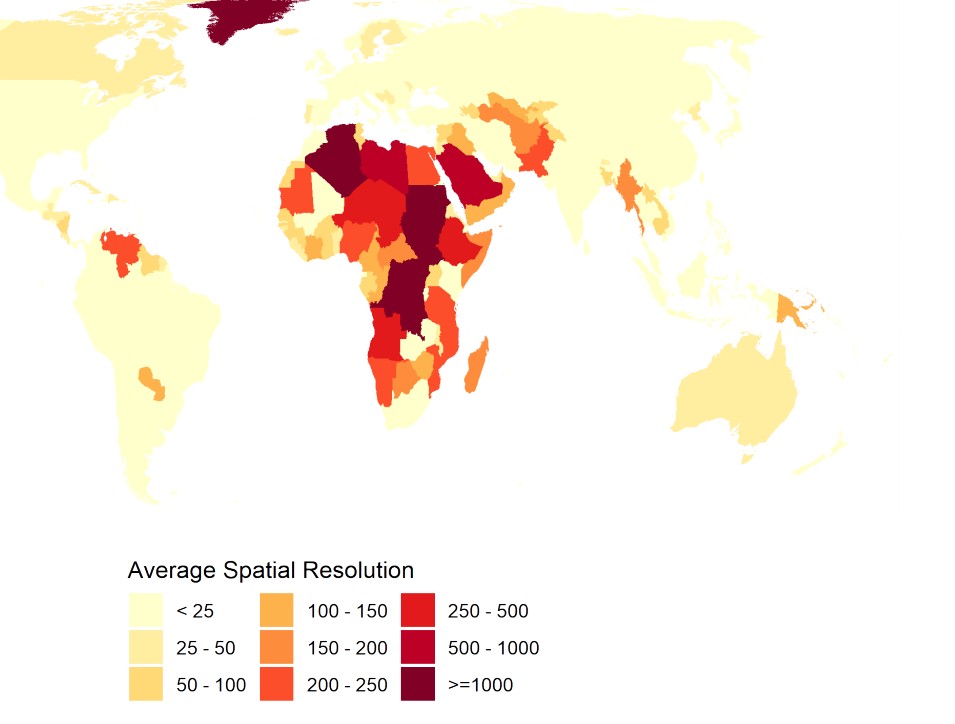

Mapping Project (GRUMP) Alpha version (Balk et al., 2006). Likewise, the Gridded Livestock of the World v2.0 includes rural population density in 2006 (GRUMP) along with other predictors such as precipitation (Hijmans et al., 2005) and a modeled travel time to places with 50,000 inhabitants circa 2000 (Nelson, 2008). Anderson et al. (2015) find variation in their examination of global data products of cropping systems models. For livestock, we transform the 5 major livestock into international values from livestock products (namely, meat, milk, eggs, honey and wool). The forest (non-wood products, wood-products) components rely on a remote sensing model to estimate forest loss. With regards to the non-timber values, limitations from the sources present two challenges. The estimates use simple averages from the literature that accordingly assume a property of uniformity in the value of a hectare of forest as similar across the world and the sample of forests with literature drawn for the study is representative of the world (Siikamäki et al., 2015). The fishing model relies on proximity and association with ports or water bodies.[17] Finally, since we do not incorporate any information on hunting, the result is an even distribution across units and time.

Another source of uncertainty is the geographic distribution of the components. Ideally, we would use sub-national prices, however it was not feasible. So the results do not reflect this occurrence, and there is a potential mis-representation of administrative units with high variation of prices due to the heterogeneity of distinct urban and rural areas.

---

[17]The freshwater case does not account for any variation, whereas the marine port locations incorporate variation on vessel holds.

### 3.3 Validation

A true validation of the predictive accuracy of this model involves data collection and construction of agricultural gross regional product in different pixels and testing those independent observations against the predicted values. The regional production data are, however, generally constructed at the administrative level rather than the pixels, so validation would have to be done on an aggregation of model predictions. Few countries provide the required data to assess the prediction accuracy to examine the internal validation of the disaggregation efficiency and the data collection would be extremely costly and time-consuming. An

evaluation of prediction accuracy requires input data at a local level, which is not available for all countries.

Multiple geographic levels of AgGDP exist for the case of Brazil where we conducted a pilot study and examined the validity of various methods to disaggregate AgGDP spatially including: cross-entropy, rural population-based model and spatial regression (see Thomas et al., 2019) . Administrative divisions of Brazil consist of 558 microgregions, which are further divided into 5,564 municipios. We had AgGDP data at both microregion and municipio levels. In order to test the methods, we only

390 used statistics for the 558 microregions and allocated them to gridded pixels. Then we aggregated estimated results at the pixel level to 5564 municipios and compared them with groundtruth data. Results showed that the correlation between the predictions and actual values at the municipio level was 0.91 for the cross-entropy model. Mean Absolute Deviation (MAD) and Root Mean Square Error (RMSE) were 8,249 and 18,347, respectively, while the average of the municipios-level true values was 28,739 (R\$ 1.000). The performance of spatial regression model was slightly better than the cross-entropy model,

but it can hardly generalize to the global work since for many countries we only have one number at the national level and don't have enough degree of freedom for the regression model. The naïve rural population model had a correlation value of 0.81 between the predictions and actual values at the municipio level, and MAD and RMSE were 28,744 and and 25,397, respectively. The cross-entropy model was proven to have relatively high accuracy compared to the naïve model and better flexibility to accommodate data scarcity in certain countries, and thus chosen as the model for the global AgGDP dataset.

At the global scale, since we do not have AgGDP statistics at lower administrative levels consistently, we are not able to validate estimated results by aggregating to different geographic levels like the Brazil case. In addition, due to the volume preserving pycnophylactic property of the cross-entropy model that utilizes all available data from mixed levels and ensures that the aggregated values conform to all original values, we do not have extra data for validation. All available data have been internalized by the model to improve estimation results and thus cannot serve as external validation. Nevertheless, we

compare the results from the global cross-entropy model to that from a rural population-based model at the grid level and examined their correlation, which is a similar assessment to You et al. (2014) (as mentioned, a spatial regression model at the global scale is not feasible due to insufficient degrees of freedom). We construct a proportional allocation model using rural population count following the method in Thomas et al. (2019) for the case of Brazil. We use the 2010 Gridded Population of the World version 4 from Center for International Earth Science Information Network - CIESIN - Columbia University

(2017) adjusted to the United Nation's World Population Prospects followed by including the rural area defined by the Global Human Settlement grid for 2015: namely, "Rural cluster", "Low Density Rural grid cell", or "Very low density rural grid cell" (Pesaresi and Freire, 2019). We disaggregate national or sub-national AgGDP statistics to grids in proportion to their rural

population, with each rural individual receiving an equal portion of the AgGDP. Figure 7 shows results of the rural per capita model and the cross-entropy model together. We can test the similarity of the two global maps. Following Levine et al. (2009),

we assume a normal distribution over the 2 million land pixels and perform a pairwise student t test to test the null hypothesis that both maps were identical. This test allows us to examine whether the mean difference in the corresponding pixel value from one map to another was greater than would be expected by chance alone. The t test statistic tell us that we can not reject the null hypothesis which provides some evidence of similarity between the two models using all the global pixels. However at a granular spatial level, Figure 8 shows variation in local correlation across the world. We use a Spearman correlation for

a 3 x 3 window of pixels with a focus on AgGDP areas with values above 200,000, excluding the Low Agricultural GDP/NA category where the measurement of rural population and AgGDP may have discontinuity due to modeling inaccuracies. The lack of similarity illustrates the difference in the spatial distribution of agricultural production systems that are not directly correlated with population density within a geographic level. At the granular spatial level, populated places and agricultural land use are different locations to allocate AgGDP. The rural per capita model is dependent on the input geographic level,

where average spatial resolution may vary, as well as on the quality and resolution of ancillary data like built-up area (e.g. Rubinyi et al., 2021).

## 4 Illustration of use: drought risk and water scarcity

Following previous global studies (e.g. Blankespoor et al., 2017; Rentschler et al., 2022), we present an application of the population exposed to a natural hazard. Specifically, we investigate the spatial distribution of population and agricultural

activity with regards to drought and water scarcity. These two indicators provide an illustrative example of different linkages to agricultural production. Drought highlights the linkages to crops and livestock whereas water scarcity focuses attention on the distribution of population. The global population estimates for the year 2010 are from WorldPop  and Center for International Earth Science Information Network (CIESIN), Columbia University (2018).[18] For a drought index, we calculate the Standardized Precipitation-Evapotranspiration Index (SPEI) (Vicente-Serrano et al., 2010), which measures the difference

between observed precipitation and estimated potential evapotranspiration with a 3 month interval using the base climatology of 1980 to 2019, which is implemented in R (Beguería and Vicente-Serrano, 2017) using climate data from Harris et al. (2020). Extra dry years are defined as the number of years that are less than or equal to -2.0 during the period from 2000 to 2009. Figure A1 shows the results of the SPEI. The Water Crowding Index (WCI) is a measure of water scarcity considering the local population as the annual water availability per capita (Falkenmark, 1986, 2013). Veldkamp et al. (2015) models global water

crowding index with return periods. We take the mean of any pixels of the ensemble WCI with a 10 year return period within an AgGDP pixel. Following the literature (e.g. Arnell, 2003; Alcamo et al., 2007; Kummu et al., 2010; Veldkamp et al., 2015), we categorize the WCI into four categories: Absolute is less than 500 $m^3$/capita per year; severe is less than or equal to 1000 $m^3$/capita per year; moderate is less than or equal to 1,700 $m^3$/capita per year; and low is the remainder (Figure A2). Then, we evaluate water shortage events using a threshold of 1,700 $m^3$/capita per year with a return period of 10 years.

---

[18]They use a Random Forest-based dasymetric redistribution method.

**Figure 7.** A panel map of gridded AgGDP circa 2010 from the Cross-Entropy model (top) and from the rural per capita population model (bottow). Source: Authors' calculation (2022).

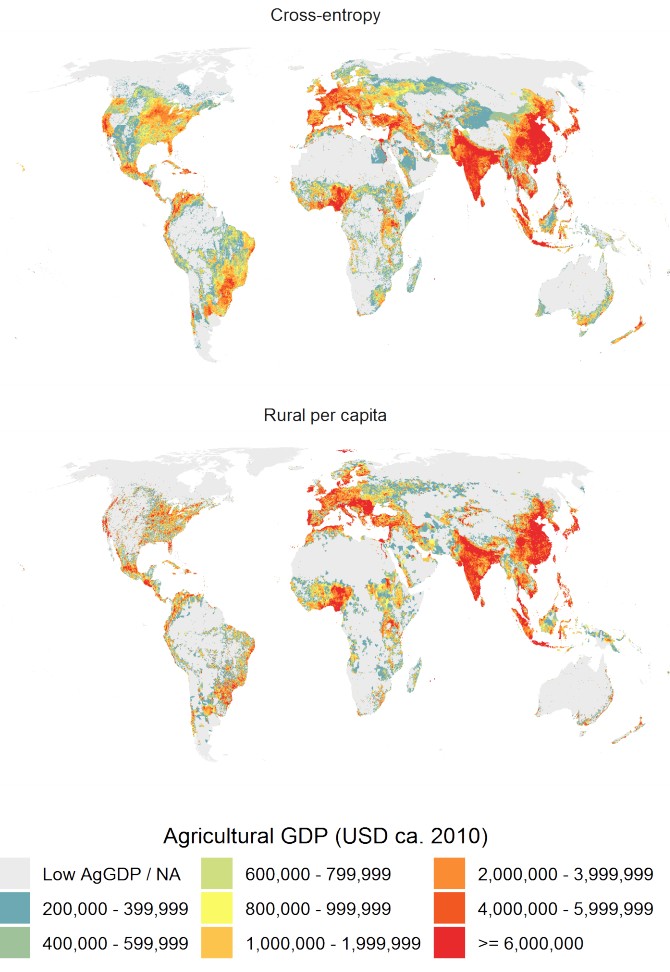

**Figure 8.** Spearman correlation in areas of AgGDP above or equal to 200,000 in the Cross-Entropy and rural per capita models. Source: Authors' calculation (2022).

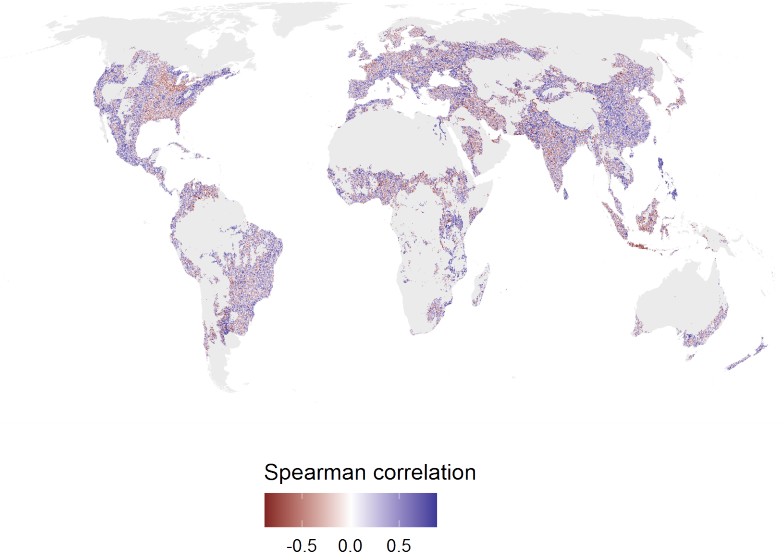

Spearman correlation

-0.5   0.0   0.5

The exposure to drought is not uniform across the world. Across the world, the group of high income countries has less population and AgGDP exposed to drought in each number of years with extremely dry compared to the countries in other income categories (Figure 9). Areas that are exposed to at least one extreme drought from 2000 to 2009 account for an estimated AgGDP of US$432 billion and a population of 1.2 billion. The top ten countries in total AgGDP exposure include the large economies in the agriculture sector such as China, India, the United States and Russian Federation (Table B1). However, other

countries have a high share of their AgGDP exposed to an extreme drought (Table B2). The top 10 countries in 2010 population exposed to dry areas include countries with the largest economies in the agriculture sector as noted above, but the list includes countries such as the Democratic Republic of Congo, Tanzania and Uganda (Table B3).

Across the world, high income countries have less population and AgGDP in areas of absolute or severe categories of the Water Crowding Index compared to countries in other income categories (Figure 10). The top ten countries of AgGDP exposed

to the Water Crowding Index include large economies in the agriculture sector such as China, India, Pakistan, Indonesia and Nigeria (Table B4). However, several countries have a high share of their AgGDP exposed to the Water Crowding Index (Table B5). The top 10 countries in 2010 population exposed to dry areas include countries with the largest economies in the agriculture sector as noted above, but the list includes countries such as Bangladesh, the Arab Republic of Egypt and Mexico (Table B6).

**Figure 9.** The total exposure of AgGDP [A] and population [B] aggregated from areas with at least one extreme drought from 2000 to 2009 measured by a 3 month SPEI. Sources: WorldPop and Center for International Earth Science Information Network (CIESIN), Columbia University (2018); World Bank (2019); Authors' calculation (2022).

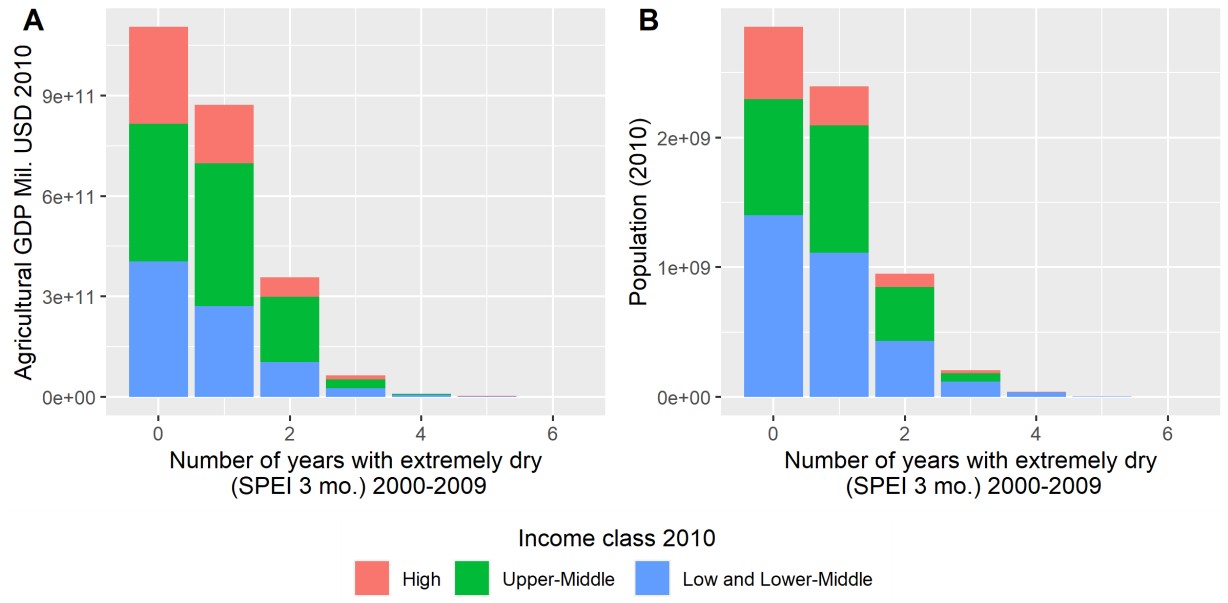

## 5   Conclusions

A globally consistent dataset on local estimates of AgGDP could benefit research and policymaking in a wide range of areas related to nature conservation, economic development, and disaster management. However, such data have been missing. In this paper, we made the first attempt to create a novel global dataset that disaggregates the national and regional accounts of the agriculture sector into 5-arcminute grids using cross-entropy optimization based on ancillary data of satellite-derived products. The gridded data format provides flexibility when the map is integrated with other data sources. It can be aggregated to various levels using administrative boundaries or other boundaries of interest, such as natural hazard zones. Since most interventions are geographically targeted, this dataset will provide important information on local variations in agricultural production and help identify places of policy interest. We illustrate the usage of this dataset through an exposure analysis of agriculture production to drought risk and water scarcity, and examine uneven natural hazard exposure across the world on US$ 432 billion of AgGDP and 1.2 billion people. With increasing frequency and severity of natural hazards such as floods, droughts, and cyclones, socio-economic estimates at the local level play a more and more important role in informing the preparations of disaster response.

These data are the result of data collection and collaboration across multiple entities to ensure the most current and widest coverage. However, persistent challenges to data collection remain, including limited geographic levels and temporal lags with

**Figure 10.** Total AgGDP [A] and population [B] by mean Water Crowding Index, where Absolute is less than 500 m³/capita per year, severe is less than or equal to 1000 m³/capita per year, moderate is less than or equal to 1700 m³/capita per year and low is the remainder. Sources: Veldkamp et al. (2015); World Bank (2019); Authors' calculation (2022).

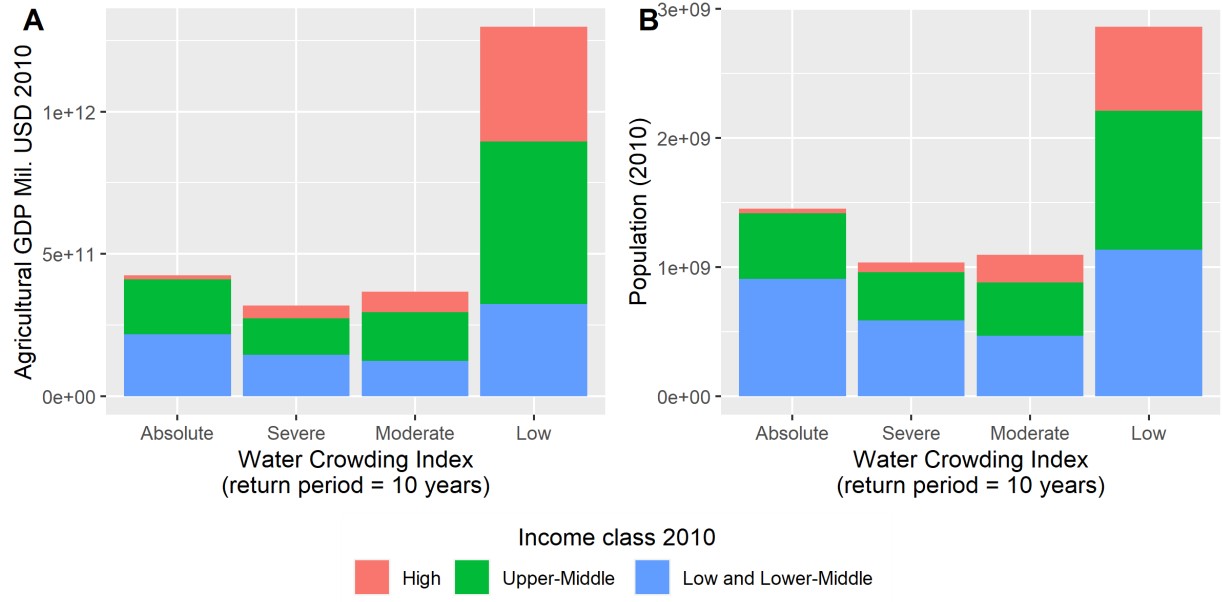

low frequencies. Also, the reference year and spatial resolution of the local AgGDP estimates are limited to the contemporaneous availability of the economic statistics and components such as the crop production model. We often have to consider the fitness-for-use while considering the accuracy; the model has higher ASR in areas where we have little data, however these same areas may benefit from the availability of these estimates to inform policy. Predictions are dependent on the availability and quality of the training data on which the model is based and the modeling process is flexible to update individual countries as the data are available.

In the near future, we hope to update this dataset as the currency and number of countries with sub-national data increase along with updated data for different agricultural components. We have learned that the main input for our crop component, SPAM, now includes data for 2017 in Sub-Saharan Africa and is in the process of producing a global crop map for 2020. Additionally, the FAO livestock distribution maps for our livestock component have been updated to include a greater variety of animal types for the more recent year of 2015. We also intend to utilize annually updated satellite imagery from MODIS Land Cover and ESA-CCI in order to calculate more recent data for the forestry and fishery sectors. In future work, we will also make the necessary adjustments to include fuel wood production and exclude trees that are cut down for plantation replanting and not used for further timber production in the calculation of forestry sector GDP.

*Data availability.* These data are available at the World Bank's Development Data Hub under http://www.doi.org/10.57966/0j71-8d56 (IF-

490   PRI and World Bank, 2022).

**Figure A1.** The number of years with at least one extreme drought from 2000 to 2009 measured by a 3 month SPEI. Sources: Harris et al. (2020); Beguería and Vicente-Serrano (2017); Authors' calculation (2022)

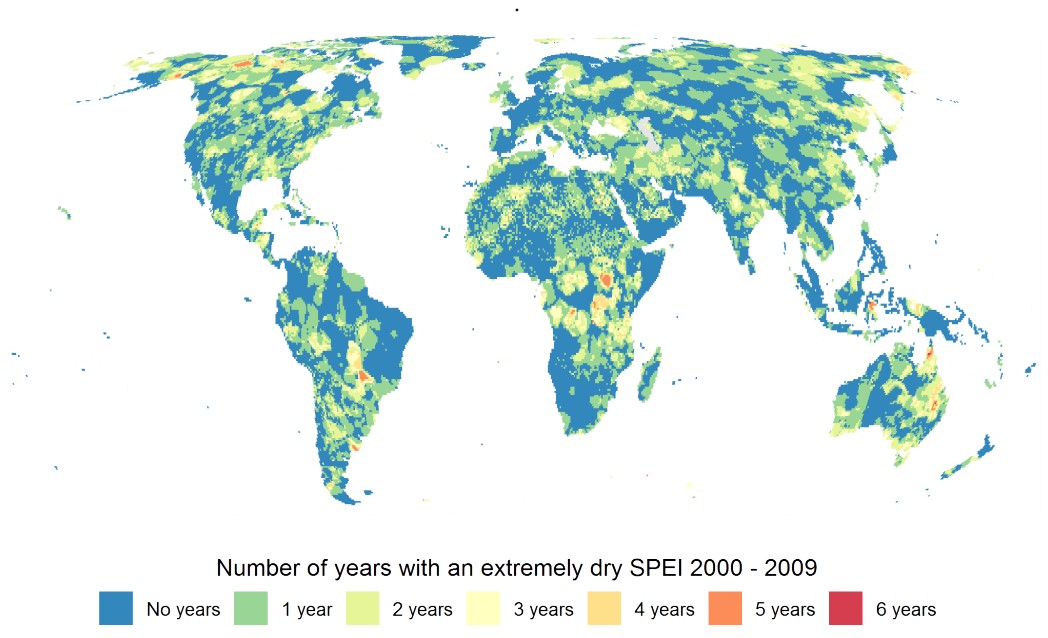

**Figure A2.** Water Scarcity Index categories with a return period of 10 years. Sources: Veldkamp et al. (2015); Authors' calculation (2022)

.

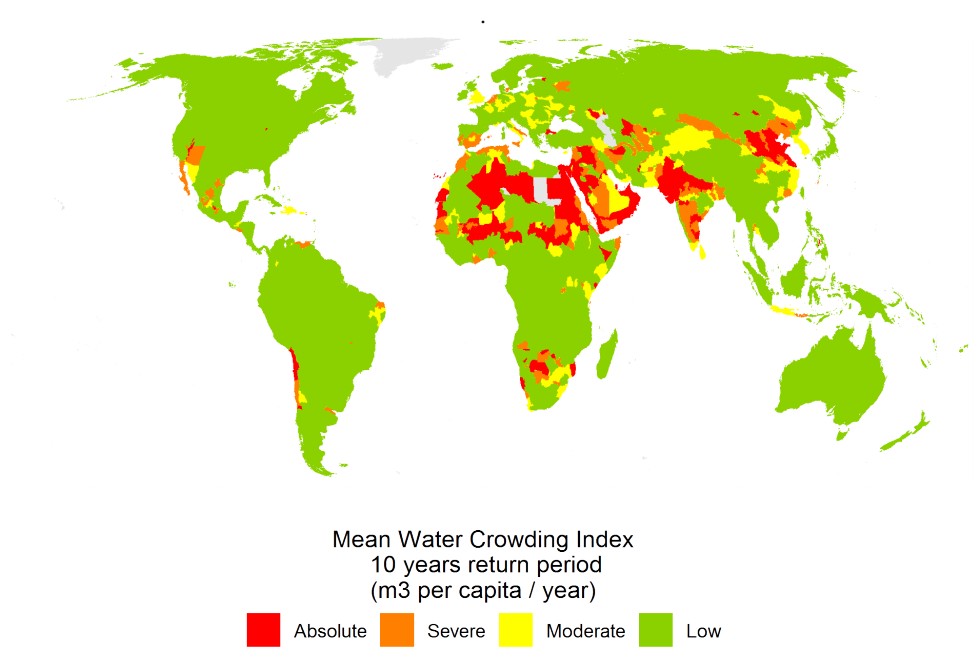

## Appendix B: Tables

**Table B1.** Top 10 countries of largest total Agricultural GDP (millions of USD) exposed to dry areas with share of Agricultural GDP and Population (thousands)

| Rank | Country | Ag GDP | Share of Ag GDP | Pop (2010) |
|---|---|---|---|---|
| 1 | China | 146,000 | 0.26 | 323,000 |
| 2 | India | 60,600 | 0.22 | 255,000 |
| 3 | United States | 21,800 | 0.14 | 69,100 |
| 4 | Russian Federation | 14,300 | 0.26 | 27,100 |
| 5 | Iran, Islamic Rep. | 13,400 | 0.44 | 40,600 |
| 6 | Brazil | 12,600 | 0.14 | 9,230 |
| 6 | Pakistan | 12,600 | 0.28 | 42,600 |
| 7 | Australia | 10,900 | 0.44 | 6,130 |
| 8 | Italy | 6,560 | 0.17 | 7,120 |
| 9 | Canada | 5,540 | 0.25 | 5,000 |

**Table B2.** Top 10 countries of largest share of Agricultural GDP exposed to dry areas with Agricultural GDP (millions of USD) and Population (thousands)

| Rank | Country | Share of Ag GDP | Ag GDP | Pop (2010) |
|---|---|---|---|---|
| 1 | Rwanda | 1.00 | 1670 | 9850 |
| 1 | Saint Vincent and the Grenadines | 1.00 | 11.6 | 29.9 |
| 1 | Micronesia, Federated States of | 1.00 | < 1 | < 1 |
| 2 | Burundi | 0.97 | 732 | 8320 |
| 3 | Brunei Darussalam | 0.91 | 99.3 | 92.8 |
| 4 | West Bank and Gaza | 0.85 | 543 | 2770 |
| 5 | Gambia, The | 0.81 | 170 | 1420 |
| 6 | Finland | 0.79 | 4400 | 3950 |
| 7 | Belize | 0.79 | 126 | 208 |
| 8 | Jordan | 0.73 | 733 | 5400 |

**Table B3.** Top 10 countries of 2010 population (thousands) exposed to dry areas with Agricultural GDP (millions of USD) and share of Agricultural GDP

| Rank | Country | Pop (2010) | Ag GDP | Share of Ag GDP |
|------|---------|-----------|--------|-----------------|
| 1 | China | 323,000 | 146,000 | 0.26 |
| 2 | India | 255,000 | 60,600 | 0.22 |
| 3 | United States | 69,100 | 21,800 | 0.14 |
| 4 | Congo, Dem. Rep. | 45,100 | 2,780 | 0.59 |
| 5 | Pakistan | 42,600 | 12,600 | 0.28 |
| 6 | Iran, Islamic Rep. | 40,600 | 13,400 | 0.44 |
| 7 | Russian Federation | 27,100 | 14,300 | 0.26 |
| 8 | Tanzania | 23,200 | 4,140 | 0.55 |
| 9 | Uganda | 18,700 | 2,990 | 0.66 |
| 10 | Thailand | 17,400 | 4,930 | 0.15 |

**Table B4.** Top 10 countries of largest total Agricultural GDP exposed to WCI areas with Agricultural GDP (million of USD) and Population (thousands)

| Rank | Country | Ag GDP | Share of Ag GDP | Pop (2010) |
|---|---|---|---|---|
| 1 | China | 436,000 | 0.802 | 990,000 |
| 2 | India | 243,000 | 0.925 | 1,000,000 |
| 3 | Pakistan | 44,200 | 0.999 | 170,000 |
| 4 | Nigeria | 38,300 | 0.465 | 78,000 |
| 5 | Indonesia | 38,200 | 0.479 | 120,000 |
| 6 | United States of America | 37,800 | 0.247 | 65,000 |
| 7 | Turkey | 37,600 | 0.625 | 43,000 |
| 8 | Italy | 30,400 | 0.854 | 42,000 |
| 9 | Iran, Islamic Republic of | 28,100 | 0.943 | 70,000 |
| 10 | Egypt, Arab Republic of | 24,400 | 0.947 | 70,000 |

**Table B5.** Top 10 countries of largest share of Agricultural GDP in country exposed to WCI areas with Agricultural GDP (million of USD) and Population (thousands)

| Rank | Country | Ag GDP | Share of Ag GDP | Pop (2010) |
|---|---|---|---|---|
| 1 | United Arab Emirates | 1,310 | 1.000 | 3,900 |
| 1 | Cyprus | 346 | 1.000 | 610 |
| 1 | Djibouti | 28 | 1.000 | 380 |
| 1 | Dominican Republic | 2,740 | 1.000 | 6,300 |
| 1 | Gambia, The | 147 | 1.000 | 680 |
| 1 | Haiti | 1,070 | 1.000 | 5,900 |
| 1 | Israel | 3,270 | 1.000 | 5,600 |
| 1 | Jamaica | 523 | 1.000 | 1,400 |
| 1 | Jordan | 996 | 1.000 | 5,800 |
| 1 | Korea, Republic of | 14,600 | 1.000 | 31,000 |

---

[18]Additional countries exposed to WCI area with the 1.00 share of Ag GDP include: West Bank and Gaza; Cyprus; Kuwait; Gambia, The; Qatar; Hong Kong (SAR, China)

**Table B6.** Top 10 countries of 2010 population exposed to WCI areas with Agricultural GDP (million of USD) and Population (thousands)

| Rank | Country | Pop (2010) | Ag GDP | Share of Ag GDP |
|---|---|---|---|---|
| 1 | India | 1,000,000 | 243,000 | 0.925 |
| 2 | China | 990,000 | 436,000 | 0.802 |
| 3 | Pakistan | 170,000 | 44,200 | 0.999 |
| 4 | Indonesia | 120,000 | 38,200 | 0.479 |
| 5 | Bangladesh | 110,000 | 13,900 | 0.909 |
| 6 | Nigeria | 78,000 | 38,300 | 0.465 |
| 7 | Egypt, Arab Republic of | 70,000 | 24,400 | 0.947 |
| 7 | Iran, Islamic Republic of | 70,000 | 28,100 | 0.943 |
| 8 | United States of America | 65,000 | 37,800 | 0.247 |
| 9 | Mexico | 64,000 | 14,100 | 0.462 |

**Table B7.** Regional account descriptive statistics

| Country | First Year | Last Year | Number of regions | Source |
|---|---|---|---|---|
| Albania | 2012 | 2014 | 12 | EUROSTAT |
| Argentina | 2004 | 2004 | 24 | Instituto Nacional de Estadística y Censos |
| Australia | 2009 | 2011 | 8 | Australian Bureau of Statistics |
| Austria | 2012 | 2014 | 9 | EUROSTAT |
| Belarus | 2011 | 2013 | 8 | BELSTAT |
| Belgium | 2012 | 2014 | 3 | EUROSTAT |
| Bolivia | 2009 | 2011 | 9 | Instituto Nacional de Estadística |
| Brazil | 2010 | 2012 | 31 | Instituto Brasileiro de Geografia e Estatística |
| Bulgaria | 2012 | 2014 | 2 | EUROSTAT |
| Canada | 2009 | 2011 | 13 | Statistics Canada |
| Chile | 2013 | 2015 | 13 | Banco Central De Chile |
| China | 2009 | 2011 | 32 | National Bureau of Statistics China |
| Colombia | 2009 | 2011 | 32 | Departamento Administrativo Nacional de Estadística |
| Croatia | 2012 | 2014 | 3 | EUROSTAT |
| Czech Republic | 2012 | 2014 | 7 | EUROSTAT |
| Denmark | 2012 | 2014 | 5 | EUROSTAT |
| Ecuador | 2006 | 2006 | 23 | Banco Central De Ecuador |
| Estonia | 2012 | 2014 | 5 | EUROSTAT |
| Finland | 2012 | 2014 | 2 | EUROSTAT |
| France | 2012 | 2014 | 22 | EUROSTAT |
| Georgia | 2009 | 2011 | 9 | National Statistics Office of Georgia |
| Germany | 2012 | 2014 | 16 | EUROSTAT |
| Greece | 2012 | 2014 | 13 | EUROSTAT |
| Hungary | 2012 | 2014 | 3 | EUROSTAT |
| India | 2011 | 2013 | 32 | Central Statistics Office |
| Indonesia | 2009 | 2011 | 31 | INDO-DAPOER |
| Iran, Islamic Rep. | 2014 | 2014 | 28 | Iran Statistical Yearbook 1389 |
| Ireland | 2012 | 2014 | 2 | EUROSTAT |
| Italy | 2012 | 2014 | 20 | EUROSTAT |
| Japan | 2009 | 2011 | 47 | Cabinet Office Government of Japan |
| Kazakhstan | 2010 | 2012 | 15 | Agency of Statistics of the Republic of Kazakhstan |
| Kenya | 2017 | 2017 | 48 | Kenya National Bureau of Statistics and World Bank |
| Korea, Rep. | 2009 | 2011 | 15 | Korean Statistical Information Services |

| Country | First Year | Last Year | Number of regions | Source |
|---|---|---|---|---|
| Latvia | 2012 | 2014 | 6 | EUROSTAT |
| Lithuania | 2012 | 2014 | 10 | EUROSTAT |
| Malaysia | 2010 | 2012 | 16 | Department of Statistics Malaysia |
| Mali | 2009 | 2009 | 9 | Cellule d'Analyse et de Prospective |
| Malta | 2012 | 2014 | 2 | EUROSTAT |
| Mexico | 2009 | 2011 | 32 | Instituto Nacional de Estadística y Geografía |
| Mongolia | 2015 | 2017 | 23 | Mongolian Statistical Information Service |
| Morocco | 2005 | 2007 | 7 | Ministry of Finance |
| Nepal | 2019 | 2019 | 7 | Central Bureau of Statistics Nepal |
| Netherlands | 2012 | 2014 | 12 | EUROSTAT |
| New Zealand | 2009 | 2011 | 14 | Statistics New Zealand |
| North Macedonia | 2012 | 2014 | 8 | EUROSTAT |
| Norway | 2012 | 2014 | 19 | EUROSTAT |
| Panama | 2009 | 2011 | 9 | Instituto Nacional de Estadística y Censo |
| Peru | 2009 | 2011 | 25 | Instituto Nacional de Estadistica e informatica |
| Philippines | 2009 | 2011 | 17 | Philippine Statistics Authority |
| Poland | 2012 | 2014 | 15 | EUROSTAT |
| Romania | 2012 | 2014 | 4 | EUROSTAT |
| Russian Federation | 2009 | 2011 | 82 | Mordoviastat: Federal Service of State Statistics |
| Slovak Republic | 2012 | 2014 | 4 | EUROSTAT |
| Slovenia | 2012 | 2014 | 2 | EUROSTAT |
| South Africa | 2009 | 2011 | 9 | Statistics South Africa |
| Spain | 2012 | 2014 | 19 | EUROSTAT |
| Sri Lanka | 2009 | 2011 | 9 | Economic and Social Statistics of Sri Lanka |
| Sweden | 2012 | 2014 | 3 | EUROSTAT |
| Switzerland | 2009 | 2011 | 25 | Federal Statistical Office of Switzerland |
| Thailand | 2009 | 2011 | 76 | Office of the National Economic and Social Development Board |
| Türkiye | 2009 | 2011 | 81 | Turkish Statistical Institute |
| Ukraine | 2010 | 2012 | 25 | State Statistics Service of Ukraine |
| United Kingdom | 2012 | 2014 | 4 | EUROSTAT |
| United States | 2009 | 2011 | 51 | Bureau of Economic Analysis |
| Uruguay | 2008 | 2008 | 19 | Instituto Nacional de Estadistica |
| Vietnam | 2009 | 2011 | 64 | General Statistics Office of Viet Nam |
| Zambia | 2015 | 2015 | 9 | Central Statistics Office of Zambia |

**Table B8.** Share of priors in territory

| Country or political area | Crop | Livestock | Timber | Non Timber | Fish |
|---|---|---|---|---|---|
| Afghanistan | 0.785 | 0.210 | 0.004 | 0.001 | 0.000 |
| Albania | 0.442 | 0.507 | 0.001 | 0.045 | 0.004 |
| Algeria | 0.638 | 0.344 | 0.012 | 0.000 | 0.006 |
| Andorra | 0.002 | 0.241 | 0.671 | 0.085 | 0.001 |
| Angola | 0.976 | 0.012 | 0.003 | 0.002 | 0.007 |
| Antigua and Barbuda | 0.461 | 0.539 | 0.000 | 0.000 | 0.000 |
| Argentina | 0.577 | 0.364 | 0.037 | 0.010 | 0.012 |
| Armenia | 0.538 | 0.455 | 0.000 | 0.003 | 0.004 |
| Australia | 0.422 | 0.391 | 0.164 | 0.001 | 0.022 |
| Austria | 0.225 | 0.315 | 0.449 | 0.010 | 0.001 |
| Azerbaijan | 0.644 | 0.354 | 0.001 | 0.001 | 0.000 |
| Bahamas, The | 0.568 | 0.350 | 0.069 | 0.014 | 0.000 |
| Bahrain | 0.100 | 0.092 | 0.020 | 0.000 | 0.788 |
| Bangladesh | 0.692 | 0.088 | 0.057 | 0.001 | 0.162 |
| Barbados | 0.341 | 0.280 | 0.379 | 0.000 | 0.000 |
| Belarus | 0.527 | 0.439 | 0.018 | 0.014 | 0.002 |
| Belgium | 0.324 | 0.455 | 0.202 | 0.006 | 0.013 |
| Belize | 0.433 | 0.185 | 0.018 | 0.357 | 0.007 |
| Benin | 0.941 | 0.010 | 0.031 | 0.001 | 0.016 |
| Bermuda (UK) | 0.505 | 0.000 | 0.495 | 0.000 | 0.000 |
| Bhutan | 0.584 | 0.190 | 0.187 | 0.036 | 0.003 |
| Bolivia | 0.487 | 0.316 | 0.002 | 0.187 | 0.009 |
| Bosnia and Herzegovina | 0.545 | 0.293 | 0.012 | 0.138 | 0.012 |
| Botswana | 0.209 | 0.402 | 0.388 | 0.001 | 0.000 |
| Brazil | 0.514 | 0.372 | 0.077 | 0.029 | 0.008 |
| British Virgin Islands (UK) | 0.237 | 0.763 | 0.000 | 0.000 | 0.000 |
| Brunei Darussalam | 0.524 | 0.362 | 0.092 | 0.001 | 0.021 |
| Bulgaria | 0.608 | 0.293 | 0.034 | 0.058 | 0.007 |
| Burkina Faso | 0.623 | 0.266 | 0.098 | 0.000 | 0.013 |
| Burundi | 0.852 | 0.095 | 0.019 | 0.008 | 0.026 |
| Cabo Verde | 0.957 | 0.000 | 0.043 | 0.000 | 0.000 |
| Cambodia | 0.716 | 0.119 | 0.045 | 0.007 | 0.114 |
| Cameroon | 0.788 | 0.116 | 0.068 | 0.004 | 0.023 |

| Country or political area | Crop | Livestock | Timber | Non Timber | Fish |
|---|---|---|---|---|---|
| Canada | 0.359 | 0.264 | 0.337 | 0.014 | 0.026 |
| Cayman Islands (UK) | 0.646 | 0.354 | 0.000 | 0.000 | 0.000 |
| Central African Republic | 0.577 | 0.339 | 0.024 | 0.057 | 0.003 |
| Chad | 0.549 | 0.436 | 0.013 | 0.002 | 0.000 |
| Chile | 0.258 | 0.142 | 0.233 | 0.014 | 0.354 |
| China | 0.565 | 0.276 | 0.093 | 0.001 | 0.065 |
| Colombia | 0.507 | 0.368 | 0.001 | 0.101 | 0.023 |
| Comoros | 0.000 | 0.207 | 0.720 | 0.073 | 0.000 |
| Congo, Dem. Rep. | 0.707 | 0.051 | 0.004 | 0.084 | 0.154 |
| Congo, Rep. of | 0.732 | 0.181 | 0.056 | 0.001 | 0.030 |
| Costa Rica | 0.747 | 0.155 | 0.030 | 0.054 | 0.014 |
| Côte d'Ivoire | 0.867 | 0.086 | 0.045 | 0.000 | 0.002 |
| Croatia | 0.484 | 0.283 | 0.194 | 0.031 | 0.008 |
| Cuba | 0.896 | 0.056 | 0.000 | 0.046 | 0.002 |
| Cyprus | 0.344 | 0.539 | 0.096 | 0.000 | 0.020 |
| Czech Republic | 0.277 | 0.323 | 0.367 | 0.026 | 0.007 |
| Denmark | 0.126 | 0.306 | 0.061 | 0.002 | 0.506 |
| Djibouti | 0.199 | 0.801 | 0.000 | 0.000 | 0.000 |
| Dominica | 0.701 | 0.106 | 0.000 | 0.193 | 0.000 |
| Dominican Republic | 0.607 | 0.282 | 0.000 | 0.105 | 0.006 |
| Ecuador | 0.423 | 0.292 | 0.076 | 0.070 | 0.139 |
| Egypt, Arab Rep. | 0.567 | 0.367 | 0.011 | 0.000 | 0.055 |
| El Salvador | 0.892 | 0.075 | 0.008 | 0.024 | 0.002 |
| Equatorial Guinea | 0.544 | 0.109 | 0.320 | 0.006 | 0.021 |
| Eritrea | 0.363 | 0.636 | 0.001 | 0.000 | 0.000 |
| Estonia | 0.226 | 0.339 | 0.375 | 0.031 | 0.028 |
| Ethiopia | 0.604 | 0.331 | 0.060 | 0.002 | 0.003 |
| Fiji | 0.458 | 0.335 | 0.153 | 0.040 | 0.014 |
| Finland | 0.090 | 0.175 | 0.692 | 0.009 | 0.033 |
| France | 0.476 | 0.350 | 0.157 | 0.005 | 0.012 |
| Gabon | 0.348 | 0.082 | 0.508 | 0.001 | 0.061 |
| Gambia, The | 0.605 | 0.374 | 0.019 | 0.000 | 0.001 |
| Georgia | 0.441 | 0.509 | 0.030 | 0.019 | 0.001 |

| Country or political area | Crop | Livestock | Timber | Non Timber | Fish |
|---|---|---|---|---|---|
| Germany | 0.254 | 0.417 | 0.307 | 0.016 | 0.006 |
| Ghana | 0.829 | 0.054 | 0.094 | 0.000 | 0.023 |
| Gibraltar (UK) | 0.011 | 0.013 | 0.004 | 0.000 | 0.972 |
| Greece | 0.711 | 0.216 | 0.046 | 0.007 | 0.020 |
| Grenada | 0.723 | 0.247 | 0.030 | 0.000 | 0.000 |
| Guatemala | 0.642 | 0.108 | 0.002 | 0.245 | 0.003 |
| Guinea | 0.747 | 0.141 | 0.079 | 0.023 | 0.011 |
| Guinea-Bissau | 0.559 | 0.414 | 0.024 | 0.002 | 0.001 |
| Guyana | 0.696 | 0.031 | 0.007 | 0.257 | 0.009 |
| Haiti | 0.794 | 0.123 | 0.000 | 0.082 | 0.000 |
| Honduras | 0.491 | 0.205 | 0.002 | 0.293 | 0.010 |
| Hong Kong (SAR, China) | 0.000 | 0.280 | 0.674 | 0.000 | 0.046 |
| Hungary | 0.537 | 0.335 | 0.111 | 0.010 | 0.006 |
| Iceland | 0.006 | 0.068 | 0.009 | 0.000 | 0.916 |
| India | 0.683 | 0.219 | 0.073 | 0.001 | 0.024 |
| Indonesia | 0.658 | 0.158 | 0.109 | 0.003 | 0.073 |
| Iran, Islamic Rep. | 0.605 | 0.364 | 0.021 | 0.000 | 0.011 |
| Iraq | 0.564 | 0.411 | 0.003 | 0.000 | 0.023 |
| Ireland | 0.091 | 0.787 | 0.093 | 0.000 | 0.028 |
| Israel | 0.523 | 0.395 | 0.074 | 0.000 | 0.009 |
| Italy | 0.463 | 0.310 | 0.207 | 0.006 | 0.014 |
| Jamaica | 0.866 | 0.059 | 0.000 | 0.066 | 0.008 |
| Japan | 0.430 | 0.244 | 0.259 | 0.001 | 0.065 |
| Jordan | 0.434 | 0.485 | 0.075 | 0.000 | 0.005 |
| Kazakhstan | 0.500 | 0.485 | 0.012 | 0.000 | 0.003 |
| Kenya | 0.555 | 0.380 | 0.031 | 0.001 | 0.033 |
| Kiribati | 0.000 | 0.034 | 0.000 | 0.000 | 0.966 |
| Korea, Democratic People's Republic of | 0.635 | 0.174 | 0.036 | 0.143 | 0.012 |
| Korea, Rep. | 0.500 | 0.241 | 0.156 | 0.001 | 0.102 |
| Kosovo | 0.631 | 0.315 | 0.024 | 0.029 | 0.002 |
| Kuwait | 0.263 | 0.426 | 0.291 | 0.000 | 0.020 |
| Kyrgyz Republic | 0.467 | 0.529 | 0.002 | 0.001 | 0.000 |
| Lao People's Democratic Republic | 0.735 | 0.164 | 0.046 | 0.017 | 0.038 |
| Latvia | 0.228 | 0.215 | 0.497 | 0.035 | 0.024 |

| Country or political area | Crop | Livestock | Timber | Non Timber | Fish |
|---|---|---|---|---|---|
| Lebanon | 0.667 | 0.257 | 0.072 | 0.000 | 0.003 |
| Lesotho | 0.303 | 0.597 | 0.092 | 0.007 | 0.001 |
| Liberia | 0.663 | 0.086 | 0.203 | 0.043 | 0.005 |
| Libya | 0.579 | 0.278 | 0.026 | 0.000 | 0.118 |
| Liechtenstein | 0.013 | 0.318 | 0.631 | 0.038 | 0.000 |
| Lithuania | 0.385 | 0.353 | 0.202 | 0.033 | 0.027 |
| Luxembourg | 0.161 | 0.493 | 0.338 | 0.008 | 0.000 |
| Macedonia, FYR | 0.681 | 0.255 | 0.002 | 0.061 | 0.001 |
| Madagascar | 0.498 | 0.431 | 0.055 | 0.005 | 0.011 |
| Malawi | 0.877 | 0.080 | 0.005 | 0.003 | 0.036 |
| Malaysia | 0.672 | 0.143 | 0.162 | 0.001 | 0.023 |
| Maldives | 0.982 | 0.018 | 0.000 | 0.000 | 0.000 |
| Mali | 0.440 | 0.452 | 0.059 | 0.000 | 0.049 |
| Malta | 0.374 | 0.482 | 0.069 | 0.000 | 0.075 |
| Mauritania | 0.226 | 0.771 | 0.003 | 0.000 | 0.001 |
| Mauritius | 0.858 | 0.000 | 0.077 | 0.000 | 0.065 |
| Mexico | 0.433 | 0.404 | 0.057 | 0.072 | 0.034 |
| Micronesia, Federated States of | 0.962 | 0.038 | 0.000 | 0.000 | 0.000 |
| Moldova | 0.630 | 0.319 | 0.001 | 0.045 | 0.005 |
| Monaco | 0.024 | 0.024 | 0.903 | 0.046 | 0.003 |
| Mongolia | 0.230 | 0.739 | 0.027 | 0.001 | 0.003 |
| Montenegro | 0.566 | 0.249 | 0.001 | 0.179 | 0.004 |
| Montserrat (UK) | 0.247 | 0.753 | 0.000 | 0.000 | 0.000 |
| Morocco | 0.615 | 0.351 | 0.022 | 0.000 | 0.012 |
| Mozambique | 0.645 | 0.279 | 0.048 | 0.022 | 0.005 |
| Myanmar | 0.805 | 0.108 | 0.016 | 0.010 | 0.061 |
| Namibia | 0.374 | 0.511 | 0.020 | 0.000 | 0.095 |
| Nepal | 0.733 | 0.232 | 0.004 | 0.020 | 0.011 |
| Netherlands | 0.257 | 0.547 | 0.147 | 0.004 | 0.045 |
| New Caledonia (Fr.) | 0.208 | 0.429 | 0.267 | 0.095 | 0.000 |
| New Zealand | 0.124 | 0.671 | 0.157 | 0.000 | 0.048 |
| Nicaragua | 0.526 | 0.275 | 0.001 | 0.194 | 0.003 |

| Country or political area | Crop | Livestock | Timber | Non Timber | Fish |
|---|---|---|---|---|---|
| Niger | 0.657 | 0.306 | 0.025 | 0.000 | 0.012 |
| Nigeria | 0.865 | 0.094 | 0.015 | 0.001 | 0.026 |
| Norway | 0.087 | 0.292 | 0.229 | 0.008 | 0.383 |
| Oman | 0.575 | 0.290 | 0.135 | 0.000 | 0.000 |
| Pakistan | 0.477 | 0.485 | 0.027 | 0.000 | 0.011 |
| Panama | 0.211 | 0.414 | 0.001 | 0.047 | 0.327 |
| Papua New Guinea | 0.443 | 0.158 | 0.042 | 0.033 | 0.325 |
| Paraguay | 0.602 | 0.313 | 0.010 | 0.074 | 0.001 |
| Peru | 0.369 | 0.253 | 0.029 | 0.037 | 0.312 |
| Philippines | 0.492 | 0.230 | 0.009 | 0.003 | 0.266 |
| Poland | 0.368 | 0.406 | 0.190 | 0.027 | 0.008 |
| Portugal | 0.392 | 0.273 | 0.280 | 0.012 | 0.042 |
| Puerto Rico (US) | 0.324 | 0.542 | 0.050 | 0.084 | 0.000 |
| Qatar | 0.229 | 0.445 | 0.326 | 0.000 | 0.000 |
| Romania | 0.540 | 0.329 | 0.113 | 0.017 | 0.001 |
| Russian Federation | 0.378 | 0.394 | 0.086 | 0.019 | 0.122 |
| Rwanda | 0.894 | 0.069 | 0.029 | 0.003 | 0.004 |
| Saint Kitts and Nevis | 0.703 | 0.297 | 0.000 | 0.000 | 0.000 |
| Saint Lucia | 0.552 | 0.255 | 0.000 | 0.193 | 0.000 |
| Saint Vincent and the Grenadines | 0.642 | 0.239 | 0.000 | 0.119 | 0.000 |
| San Marino | 0.776 | 0.185 | 0.036 | 0.003 | 0.000 |
| São Tomé and Príncipe | 0.935 | 0.065 | 0.000 | 0.000 | 0.000 |
| Saudi Arabia | 0.548 | 0.406 | 0.039 | 0.000 | 0.007 |
| Senegal | 0.756 | 0.084 | 0.085 | 0.000 | 0.074 |
| Serbia | 0.649 | 0.291 | 0.015 | 0.040 | 0.005 |
| Seychelles | 0.922 | 0.078 | 0.000 | 0.000 | 0.000 |
| Sierra Leone | 0.785 | 0.042 | 0.137 | 0.007 | 0.030 |
| Singapore | 0.072 | 0.145 | 0.607 | 0.000 | 0.176 |
| Slovak Republic | 0.289 | 0.228 | 0.453 | 0.029 | 0.001 |
| Slovenia | 0.230 | 0.346 | 0.390 | 0.030 | 0.004 |
| Solomon Islands | 0.586 | 0.018 | 0.285 | 0.112 | 0.000 |
| Somalia | 0.113 | 0.872 | 0.008 | 0.007 | 0.000 |
| South Africa | 0.417 | 0.375 | 0.152 | 0.000 | 0.057 |

| Country or political area | Crop | Livestock | Timber | Non Timber | Fish |
|---|---|---|---|---|---|
| South Sudan | 0.873 | 0.112 | 0.007 | 0.000 | 0.008 |
| Spain | 0.531 | 0.247 | 0.152 | 0.006 | 0.064 |
| Sri Lanka | 0.758 | 0.100 | 0.071 | 0.020 | 0.052 |
| Sudan | 0.419 | 0.562 | 0.000 | 0.000 | 0.019 |
| Suriname | 0.534 | 0.255 | 0.072 | 0.115 | 0.023 |
| Swaziland | 0.618 | 0.115 | 0.265 | 0.002 | 0.001 |
| Sweden | 0.094 | 0.150 | 0.740 | 0.014 | 0.002 |
| Switzerland | 0.182 | 0.401 | 0.410 | 0.007 | 0.000 |
| Syrian Arab Republic | 0.594 | 0.374 | 0.020 | 0.001 | 0.012 |
| Tajikistan | 0.791 | 0.206 | 0.002 | 0.000 | 0.001 |
| Tanzania | 0.798 | 0.038 | 0.051 | 0.006 | 0.106 |
| Thailand | 0.744 | 0.165 | 0.062 | 0.001 | 0.028 |
| Timor-Leste | 0.661 | 0.237 | 0.006 | 0.006 | 0.089 |
| Togo | 0.657 | 0.310 | 0.021 | 0.004 | 0.008 |
| Tonga | 0.588 | 0.000 | 0.412 | 0.000 | 0.000 |
| Trinidad and Tobago | 0.306 | 0.550 | 0.073 | 0.070 | 0.000 |
| Tunisia | 0.572 | 0.336 | 0.059 | 0.000 | 0.033 |
| Türkiye | 0.656 | 0.284 | 0.053 | 0.001 | 0.007 |
| Turkmenistan | 0.451 | 0.547 | 0.000 | 0.000 | 0.001 |
| Uganda | 0.637 | 0.090 | 0.086 | 0.002 | 0.185 |
| Ukraine | 0.671 | 0.287 | 0.005 | 0.032 | 0.005 |
| United Arab Emirates | 0.504 | 0.298 | 0.197 | 0.000 | 0.000 |
| United Kingdom | 0.285 | 0.437 | 0.235 | 0.004 | 0.040 |
| United States | 0.454 | 0.302 | 0.207 | 0.014 | 0.023 |
| United States | 0.954 | 0.036 | 0.000 | 0.000 | 0.010 |
| Uruguay | 0.239 | 0.295 | 0.076 | 0.003 | 0.388 |
| Uzbekistan | 0.758 | 0.240 | 0.001 | 0.000 | 0.001 |
| Vanuatu | 0.912 | 0.035 | 0.020 | 0.032 | 0.000 |
| Vatican City | 0.366 | 0.234 | 0.000 | 0.397 | 0.003 |
| Venezuela, Republica Bolivariana de | 0.569 | 0.364 | 0.015 | 0.028 | 0.025 |
| Vietnam | 0.599 | 0.259 | 0.044 | 0.003 | 0.095 |
| West Bank and Gaza | 0.231 | 0.752 | 0.015 | 0.000 | 0.002 |
| Yemen, Rep. | 0.516 | 0.472 | 0.012 | 0.000 | 0.000 |
| Zambia | 0.517 | 0.169 | 0.233 | 0.006 | 0.075 |
| Zimbabwe | 0.517 | 0.351 | 0.105 | 0.013 | 0.013 |

*Author contributions.* BB, LY and TT framed the work and designed the modelling framework. YR, UW, and EK carried them out. UW and EK developed the model code and performed the simulations. YR, UW, and BB prepared the input datasets for the model. BB applied and validated the data. YR and BB prepared the manuscript with contributions from all co-authors.

*Competing interests.* The authors declare no competing interests are present.

*Disclaimer.* The findings, interpretations and conclusions expressed in this paper are entirely those of the authors. They do not necessarily represent the views of the World Bank and its affiliated organizations, or those of the Executive Directors of the World Bank or the government they represent. The maps displayed in this paper are for reference only. The boundaries, colors, denominations and any other information shown on these maps do not imply, on the part of the World Bank Group any judgment on the legal status of any territory, or any endorsement or acceptance of such boundaries.

*Acknowledgements.* The authors would like to thank the following people for discussions and prior reviews: Claudia Berg (World Bank), Gero Carletto (World Bank), Uwe Deichmann (World Bank), Rashmin Gunasekera (World Bank), Barbro Hexeberg (World Bank), Glenn-Marie Lange (World Bank), Michael Lokshin (World Bank), Eric Roland Metreau (World Bank), Jose Pablo Valdes Martinez (World Bank), Tim Robinson (FAO), Steven Rubinyi (World Bank), Juha Siikamäki (IUCN), Ben Stewart (World Bank), Jeffrey R. Vincent (Duke University). We appreciate the use of non-timber value data from Juha Siikamäki and ports data from Gilles Hosch. The authors would like to thank the participants of conferences including: American Association of Geographers Annual Meeting 2019 in Washington, DC, 3-7 April 2019; United Nations Economic Commission for Europe Workshop on Data Integration: Realising the Potential of Statistical and Geospatial Data in Belgrade, Serbia, 21-23 May 2019; International Institute for Applied Systems Analysis seminar in Laxenburg, Austria, 27 September 2019; IFPRI RISE Workshop in Washington, DC, 19 November 2019; and the Committee for the Coordination of Statistical Activities and United Nations Geospatial Network Joint virtual Workshop on the Integration between Geospatial and Statistical Information, 28 April 2021. We appreciate the support of the World Bank Strategic Research Program on Big Data.

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

Glossary:Livestock_unit_(LSU), 2018.

Falkenmark, M.: Fresh water: Time for a modified approach, Ambio, pp. 192–200, 1986.

Falkenmark, M.: Growing water scarcity in agriculture: future challenge to global water security, Philosophical Transactions of the Royal Society A: Mathematical, Physical and Engineering Sciences, 371, 20120 410, 2013.

FAO: Fishery and aquaculture statistics 2009, https://www.fao.org/publications/card/en/c/1030779b-3733-5f5d-b3e4-0e779b94e498/, 2009.
FAO: World Food and Agriculture Statistical yearbook, 2013.

FAO: Global Administrative Unit Layers (GAUL) Dataset, http://www.fao.org/geonetwork/srv/en/metadata.show%3Fid=12691, 2015.

FAO: FAOSTAT Database, http://faostat.fao.org/site/291/default.aspx, 2016.

FAO: Forests and poverty reduction, http://www.fao.org/forestry/livelihoods/en/, 2018.

FAO: World Food and Agriculture Statistical pocketbook, 2019.
FAO: FAOSTAT Database, http://www.fao.org/faostat, 2020.

FAO: Impact of disasters and crises on agriculture and food security, 2021, 2021.

Friedl, M. A., Sulla-Menashe, D., Tan, B., Schneider, A., Ramankutty, N., Sibley, A., and Huang, X.: MODIS Collection 5 global land cover: Algorithm refinements and characterization of new datasets, Remote sensing of Environment, 114, 168–182, 2010.

Ghazoul, J. and Evans, J.: Sustainable Forest Management | Causes of Deforestation and Forest Fragmentation, in: Encyclopedia of Forest
Sciences, edited by Burley, J., pp. 1367–1375, Elsevier, Oxford, https://doi.org/https://doi.org/10.1016/B0-12-145160-7/00018-1, 2004.

Ghosh, T., L Powell, R., D Elvidge, C., E Baugh, K., C Sutton, P., and Anderson, S.: Shedding light on the global distribution of economic activity, The Open Geography Journal, 3, 2010.

Gibson, J., Olivia, S., Boe-Gibson, G., and Li, C.: Which night lights data should we use in economics, and where?, Journal of Development Economics, 149, 102 602, 2021.
Gilbert, M., Nicolas, G., Cinardi, G., Van Boeckel, T. P., Vanwambeke, S. O., Wint, G. W., and Robinson, T. P.: Global distribution data for cattle, buffaloes, horses, sheep, goats, pigs, chickens and ducks in 2010, Scientific data, 5, 1–11, 2018.

Goldblatt, R., Heilmann, K., and Vaizman, Y.: Can medium-resolution satellite imagery measure economic activity at small geographies? Evidence from Landsat in Vietnam, World Bank Policy Research Working Paper 9088, World Bank, Washington, DC, 2019.

Gollin, D., Lagakos, D., and Waugh, M. E.: The agricultural productivity gap, The Quarterly Journal of Economics, 129, 939–993, 2014.
Griscom, B. W., Adams, J., Ellis, P. W., Houghton, R. A., Lomax, G., Miteva, D. A., Schlesinger, W. H., Shoch, D., Siikamäki, J. V., Smith, P., et al.: Natural climate solutions, Proceedings of the National Academy of Sciences, 114, 11 645–11 650, 2017.

Gunasekera, R., Ishizawa, O., Aubrecht, C., Blankespoor, B., Murray, S., Pomonis, A., and Daniell, J.: Developing an adaptive global exposure model to support the generation of country disaster risk profiles, Earth-Science Reviews, 150, 594–608, 2015.

Gunasekera, R., Daniell, J., Pomonis, A., Arias, R., Ishizawa, O., and Stone, H.: Methodology Note on the Global RApid post-disaster Damage Estimation (GRADE) approach, World Bank and GFDRR Technical Report, World Bank and GFDRR, Washington, DC, 2018.

Harris, I., Osborn, T. J., Jones, P., and Lister, D.: Version 4 of the CRU TS monthly high-resolution gridded multivariate climate dataset, Scientific data, 7, 1–18, 2020.

Henderson, J. V., Storeygard, A., and Weil, D. N.: Measuring economic growth from outer space, American economic review, 102, 994–1028, 2012.

Herrero, M. and Thornton, P. K.: Livestock and global change: emerging issues for sustainable food systems, Proceedings of the National Academy of Sciences, 110, 20 878–20 881, 2013.

Hijmans, R. J., Cameron, S. E., Parra, J. L., Jones, P. G., and Jarvis, A.: Very high resolution interpolated climate surfaces for global land areas, International Journal of Climatology: A Journal of the Royal Meteorological Society, 25, 1965–1978, 2005.

Hosch, G., Soule, B., Schofield, M., Thomas, T., Kilgour, C., and Huntington, T.: Any Port in a Storm: Vessel Activity and the Risk of IUU-Caught Fish Passing through the World's Most Important Fishing Ports, Journal of Ocean and Coastal Economics, 6, 1, 2019.

Hossain, M. K., Alam, M. K., Miah, M. D., et al.: Forest restoration and rehabilitation in Bangladesh, Keep Asia Green, 3, 21–66, 2008.

IFPRI and World Bank: Global Gridded Agricultural Gross Domestic Product (AgGDP) [dataset], https://doi.org/10.57966/0j71-8d56, 2022.

Joglekar, A. K. B., Wood-Sichra, U., and Pardey, P. G.: Pixelating crop production: Consequences of methodological choices, PLOS ONE, 14, e0212 281, https://doi.org/10.1371/journal.pone.0212281, 2019.

Kelleher, K., Willmann, R., and Arnason, R.: The sunken billions: the economic justification for fisheries reform, The World Bank and FAO, 2009.

Kummu, M., Ward, P. J., de Moel, H., and Varis, O.: Is physical water scarcity a new phenomenon? Global assessment of water shortage over the last two millennia, Environmental Research Letters, 5, 034 006, 2010.

Kummu, M., Taka, M., and Guillaume, J. H.: Gridded global datasets for gross domestic product and Human Development Index over 1990–2015, Scientific data, 5, 180 004, 2018.

Lamarche, C., Santoro, M., Bontemps, S., d'Andrimont, R., Radoux, J., Giustarini, L., Brockmann, C., Wevers, J., Defourny, P., and Arino, O.: Compilation and validation of SAR and optical data products for a complete and global map of inland/ocean water tailored to the climate modeling community, Remote Sensing, 9, 36, 2017.

Lebedys, A. and Li, Y.: Contribution of the forestry sector to national economies, 1990-2011, Forest Finance Working Paper (FAO) eng no. 09, 2014.

Lehmann, R. and Wohlrabe, K.: Forecasting GDP at the regional level with many predictors, German Economic Review, 16, 226–254, 2015.

Levine, R. S., Yorita, K. L., Walsh, M. C., and Reynolds, M. G.: A method for statistically comparing spatial distribution maps, International Journal of Health Geographics, 8, 1–7, 2009.

Leyk, S., Gaughan, A. E., Adamo, S. B., de Sherbinin, A., Balk, D., Freire, S., Rose, A., Stevens, F. R., Blankespoor, B., Frye, C., Comenetz, J., Sorichetta, A., MacManus, K., Pistolesi, L., Levy, M., Tatem, A. J., and Pesaresi, M.: The spatial allocation of population: A review of large-scale gridded population data products and their fitness for use, Earth System Science Data, 11, 2019.

Li, T., Pullar, D., Corcoran, J., and Stimson, R.: A comparison of spatial disaggregation techniques as applied to population estimation for South East Queensland (SEQ), Australia, Applied GIS, 3, 1–16, 2007.

Li, X., Zhou, Y., Zhao, M., and Zhao, X.: A harmonized global nighttime light dataset 1992–2018, Scientific data, 7, 1–9, 2020.

Luijten, J.: A systematic method for generating land use patterns using stochastic rules and basic landscape characteristics: results for a Colombian hillside watershed, Agriculture, ecosystems & environment, 95, 427–441, 2003.

Mathiesen, À. M.: Fisheries: feeding humanity in 2030. Conference presentation at Our Ocean 2018, http://www.fao.org/fi/static-media/ADG/MathiesenOurOceanConference2018.pdf, 2018.

Murakami, D. and Yamagata, Y.: Estimation of gridded population and GDP scenarios with spatially explicit statistical downscaling, Sus-

630 tainability, 11, 2106, 2019.

Murthy, C., Laxman, B., and Sai, M. S.: Geospatial analysis of agricultural drought vulnerability using a composite index based on exposure, sensitivity and adaptive capacity, International journal of disaster risk reduction, 12, 163–171, 2015.

NASA: MODIS Collection 6 NRT Hotspot / Active Fire Detections MCD14DL, https://doi.org/DOI: 10.5067/FIRMS/MODIS-/MCD14DL.NRT.006, 2018.

National Geospatial-Intelligence Agency: World Port Index, 2019.

Nelson, A.: Travel time to major cities: A global map of Accessibility, Ispra: European Commission, 2008.

Nelson, G. C.: Introduction to the special issue on spatial analysis for agricultural economists, Agricultural Economics, 27, 197–200, 2002.

NOAA: Version 4 DMSP-OLS Global Radiance Calibrated Nighttime Lights, https://ngdc.noaa.gov/eog/dmsp/download/radcal.html, 2011.

Nordhaus, W. D.: Geography and macroeconomics: New data and new findings, Proceedings of the National Academy of Sciences, 103,

3510–3517, 2006.

Openshaw, S.: The modifiable areal unit problem, Quantitative geography: A British view, pp. 60–69, 1981.

Pesaresi, M. and Freire, S.: GHS-SMOD-GHS settlement grid, 2019.

Pratesi, M., Salvati, N., Giusti, C., and Marchetti, S.: Spatial disaggregation and small-area estimation methods for agricultural surveys: solutions and perspectives, Technical Report Series GO-07-2015, FAO Global Office of the Global Strategy, 2015.

Reddy, T. K. and Dutta, M.: Impact of Agricultural Inputs on Agricultural GDP in Indian Economy, Theoretical Economics Letters, 8, 1840–1853, 2018.

Rentschler, J. and Salhab, M.: People in harm's way: Flood exposure and poverty in 189 countries, World Bank Policy Research Working Paper 9447, World Bank, Washington, DC, 2020.

Rentschler, J., Salhab, M., and Jafino, B. A.: Flood exposure and poverty in 188 countries, Nature communications, 13, 3527, 2022.

Roberts, M., Blankespoor, B., Deuskar, C., and Stewart, B.: Urbanization and development: Is latin america and the caribbean different from the rest of the world?, World Bank Policy Research Working Paper 8019, World Bank, Washington, DC, 2017.

Robinson, T. P., Wint, G. W., Conchedda, G., Van Boeckel, T. P., Ercoli, V., Palamara, E., Cinardi, G., D'Aietti, L., Hay, S. I., and Gilbert, M.: Mapping the global distribution of livestock, PloS one, 9, e96 084, 2014.

Rubinstein, R. Y. and Kroese, D. P.: The cross-entropy method: a unified approach to combinatorial optimization, Monte-Carlo simulation,

and machine learning, vol. 133, Springer, 2004.

Rubinyi, S., Blankespoor, B., and Hall, J. W.: The utility of built environment geospatial data for high-resolution dasymetric global population modeling, Computers, Environment and Urban Systems, 86, 101 594, 2021.

Samberg, L. H., Gerber, J. S., Ramankutty, N., Herrero, M., and West, P. C.: Subnational distribution of average farm size and smallholder contributions to global food production, Environmental Research Letters, 11, 124 010, 2016.

Shyamsundar, P., Cohen, F., Boucher, T. M., Kroeger, T., Erbaugh, J. T., Waterfield, G., Clarke, C., Cook-Patton, S. C., Garcia, E., Juma, K., et al.: Scaling smallholder tree cover restoration across the tropics, Global Environmental Change, 76, 102 591, 2022.

Siikamäki, J., Santiago-Ávila, F. J., and Vail, P.: Global Assessment of Non'Wood Forest Ecosystem Services, Working paper, Resources for the Future, Washington, DC., 2015.

Sorrenti, S.: Non-wood forest products in international statistical systems, 2016.

Staal, S. J., Baltenweck, I., Waithaka, M., DeWolff, T., and Njoroge, L.: Location and uptake: integrated household and GIS analysis of technology adoption and land use, with application to smallholder dairy farms in Kenya, Agricultural Economics, 27, 295–315, 2002.

Stanimirova, R., Arévalo, P., Kaufmann, R. K., Maus, V., Lesiv, M., Havlík, P., and Friedl, M. A.: Sensitivity of global pasturelands to climate variation, Earth's Future, 7, 1353–1366, 2019.

Thomas, T. S., You, L., Wood-Sichra, U., Ru, Y., Blankespoor, B., and Kalvelagen, E.: Generating Gridded Agricultural Gross Domestic

Product for Brazil: A Comparison of Methodologies, World Bank Policy Research Working Paper WPS 8985, World Bank, Wasington, D.C., 2019.

Tobler, W. R.: Smooth pycnophylactic interpolation for geographical regions, Journal of the American Statistical Association, 74, 519–530, 1979.

UNDRR: Global assessment report on disaster risk reduction, United Nations Office for Disaster Risk Reduction (UNDRR), 2019.

UNISDR: Global Assessment Report on Disaster Risk Reduction 2011: Revealing Risk, Redefining Development, United Nations International Strategy for Disaster Reduction, Geneva, Switzerland, https://www.preventionweb.net/english/hyogo/gar/2011/en/home/index.html, 2011.

Veldkamp, T. I., Eisner, S., Wada, Y., Aerts, J. C., and Ward, P. J.: Sensitivity of water scarcity events to ENSO-driven climate variability at the global scale, Hydrology and Earth System Sciences, 19, 4081–4098, 2015.

Vesco, P., Kovacic, M., Mistry, M., and Croicu, M.: Climate variability, crop and conflict: Exploring the impacts of spatial concentration in agricultural production, Journal of Peace Research, 58, 98–113, 2021.

Vicente-Serrano, S. M., Beguería, S., and López-Moreno, J. I.: A multiscalar drought index sensitive to global warming: the standardized precipitation evapotranspiration index, Journal of climate, 23, 1696–1718, 2010.

Wang, T. and Sun, F.: Spatially explicit global gross domestic product (GDP) data set consistent with the Shared Socioeconomic Pathways,

Earth System Science Data Discussions, pp. 1–34, 2021.

Wang, X., Sutton, P. C., and Qi, B.: Global mapping of GDP at 1 km2 using VIIRS nighttime satellite imagery, ISPRS International Journal of Geo-Information, 8, 580, 2019.

Ward, P. J., Blauhut, V., Bloemendaal, N., Daniell, J. E., de Ruiter, M. C., Duncan, M. J., Emberson, R., Jenkins, S. F., Kirschbaum, D., Kunz, M., et al.: Natural hazard risk assessments at the global scale, Natural Hazards and Earth System Sciences, 20, 1069–1096, 2020.

Wood, S., Sebastian, K., Nachtergaele, F., Nielsen, D., and Dai, A.: Spatial aspects of the design and targeting of agricultural development strategies, Environment and Production Technology Division Discussion Paper No. 44, International Food Policy Research Institute, Washington, DC., 1999.

World Bank: World Development Indicators Database (World Bank), 2019.

World Bank and UNEP: Gross Domestic Product 2010, https://preview.grid.unep.ch/, 2011.

WorldPop and Center for International Earth Science Information Network (CIESIN), Columbia University: Global High Resolution Population Denominators Project - Funded by The Bill and Melinda Gates Foundation (OPP1134076), https://doi.org/https://dx.doi.org/10.5258/SOTON/WP00647, 2018.

You, L. and Wood, S.: Spatial allocation of agricultural production using a cross-entropy approach, Environment and Production Technology Division Discussion Paper No. 126, International Food Policy Research Institute, Washington, DC., 2003.

You, L. and Wood, S.: An entropy approach to spatial disaggregation of agricultural production, Agricultural Systems, 90, 329–347, 2006.

You, L., Wood, S., Wood-Sichra, U., and Wu, W.: Generating global crop distribution maps: From census to grid, Agricultural Systems, 127, 53–60, 2014.

You, L., Wood-Sichra, U., Fritz, S., Guo, Z., See, L., and Koo, J.: Spatial production allocation model (SPAM) 2010 Version 1.0, http: //MapSPAM.info, 2018.

Yu, Q., You, L., Wood-Sichra, U., Ru, Y., Joglekar, A. K., Fritz, S., Xiong, W., Lu, M., Wu, W., and Yang, P.: A cultivated planet in 2010–Part 2: the global gridded agricultural-production maps, Earth System Science Data, 12, 3545–3572, 2020.

Zhang, Y., You, L., Lee, D., and Block, P.: Integrating climate prediction and regionalization into an agro-economic model to guide agricultural planning, Climatic Change, 158, 435–451, 2020.