# Peer review of "Estimating Local AgGDP across the World"

_Earth System Science Data, 2022_

## Community Comment (CC1)

**General comments**

There is clearly a great effort behind this manuscript, but I feel the authors should focus on the refinement of the methodology in view of producing something that can be more easily updated to account for the dynamicity of the agricultural sector. The authors generate a spatial distribution of the agricultural GDP circa 2010 but this information can hardly be useful for analysis of the risks that the agricultural production face more than a decade later given that the authors themselves highlight the dynamicity of the sector. Besides, the analysis of the exposure to drought falls short to describe the adaptive capacity that characterize many agricultural systems. The authors indicate that the results of the analysis are suitable for global, continental and regional analysis but not for local analyses and one may wonder if in the end this effort is worth doing, also considering that this information cannot be easily validated and that is not suitable to inform local planning. There are some methodological issues that would require in my view some attention: for instance the spatial analysis of the agricultural GDP that is done for the wood products; collinearity of input data. The paper does not contain a quantification of the uncertainties and does not mention the fact that agricultural GDP cannot capture well agricultural production in the informal or secondary economy. A more consolidated discussion of the limitations of this product would greatly benefit the strength of this paper.

Overall, I would welcome the publication after a revision that addresses these main points: 1) Strengthen the methodology or justify better some of the technical choices that were made; 2) Provide some quantification of the uncertainties. 3) Add some discussion on how the ever-growing release of new and better inputs data (crop maps; livestock distribution; dynamic land cover maps; more spatially-disaggregate and recent statistics) may be integrated into this product to reduce or even better to keep up with the temporal mismatch in agricultural production. Finally, one suggestion: the linkages between drought and agricultural production are less important for fisheries and wood production than for crop and livestock whereas the water crowing index is more associated with the distribution of the population, which is itself quite outdated in this analysis. My suggestion would be to remove or shorten the discussion on the exposure to drought. I understand that it was used as an example of application but in my opinion doesn't really bring much value to the discussion.

**Detailed comments**

Lines 8-9: please check this sentence, is there something missing?

Lines 15-16: no need I think to discuss the limitations of administrative values versus spatially distributed variables.

Line 19: I suggest using agricultural aspects instead of agricultural issues.

Line 24: global food exposed population: is this the overall population in low and middle-income countries?

Line 24: 600 million people are in low and middle-income countries or everywhere?

Line 42: night light detection instead of luminosity?

Line 57: nighttime lights detection is clearly associated with income distribution not necessarily with the economic and processing activities which in fact happen mostly during the day

Line 59: *"night light data may not capture the agricultural activity as it requires areas to emit light."* This is not very clear.

Line 143-144: forest production is not related to forest loss. In fact I would say the opposite. In forest managed systems, trees are cut but that not necessarily reflects in a forest loss and land cover change as implied in this assumption. Land cover and land cover change maps can provide information on deforestation but hardly on forest production. One possible solution can be to spatialize country statistics of forestry

production by maps of tree cover instead (but please see the following comment). Statistics can be derived from the FAOSTAT database Forestry production and trade which contains data on the production and trade in roundwood and in primary wood and paper products for all countries and territories in the world, collected through the Joint Forest Sector Questionnaire (JFSQ) (http://www.fao.org/forestry/statistics/80572/en/). More detailed information on wood products, including definitions, can be found at http://www.fao.org/forestry/statistics/80572/en/

Line 144-145: Also, related to forest, I wonder why you're using a different land cover than the CCI already used for the water bodies. Also, please note that a new version of the MODIS land cover has been available for several years now. MODIS 5 is discontinued. In the MODIS 6 the spurious detection of land cover change were reduced compared to V5 but not entirely removed.

Lines 190-194: is this a pro-rata adjustment which is made to agricultural priors within each country or is it a proportional adjustment based on the weight of each component? It would be useful to see a table that quantifies the implications of this adjustment such as in percentage of the total. Also, production data of the crops in SPAM are stratified by the farming system. Production in two of these farming systems (rainfed low input and rainfed subsistence farming) is mostly for home consumption. One of the limitations of the agricultural GDP is that it may poorly capture production that is not exchanged on the market. And this is valid for the artisanal fisheries which may escape official (e.g. in several West African countries) but also for livestock in pastoral systems. Wonder if there are implications with regards to the spatial distribution of the GDP in countries where low-input farming systems prevail. For instance, would it still be possible to identify hotspots of more intensive production in areas with prevailing low input systems which would be useful to target interventions? In SSA countries, it is not uncommon that agricultural GDP benefits the urban population (which owns fishing vessels and herds while living in the cities). This will not be captured by the system.

Line 247: The Water Crowding Index is not only linked to agricultural production and for instance (temporary) water scarcity has less direct impact on wood products and fisheries. It's rather associated with the distribution of the population regardless of the economic activities.

Line 255: it's unclear why night lights are used in this context.

Lines: 268-269 I think the authors refer here to specialized agricultural production. Suggest rephrasing.

Line 270: I suggest rephrasing: *it requires areas to emit light* – not that clear.

Paragraph 3.2: I'd call it Uncertainties and limitations

Paragraph 3.2.2 there is collinearity between inputs (e.g. GLW and SPAM). Additionally, these modelled data all contain agroecological information and climate variables which are also implicit in the response variable (exposure to drought).

Figure 8: would one get different results doing the same analysis at country level.

Lines 336-337: given the large uncertainties and difficulties in the validation and if the results are currently suitable for applications at global, continental or regional level wouldn't it be more efficient to stick to the old and good country values?

---

## Author Response (AR1)

ESSDD Responses:

| Comments to the author | Response | action |
|---|---|---|
| I would personally like ot see in a revised manuscript a bit of a description of how original data owners were contacted or why not. by Francesco N. Tubiello | We contacted the data producers to access the data. Specifically, we produced SPAM at IFPRI and the dataset is publicly available at www. mapspam.info/data. The gridded livestock was downloaded from the FAO website (Gridded Livestock of the World (GLW2)). The gridded non-wood forest products from Resources for the Future were obtained from Juha Siikamäki(2015). The wood forest products were estimated using data downloaded from MODIS land cover and NASA FIRMS fire. Fishery data were constructed using water bodies data downloaded from ESA CCI and port data from Gilles Hosch (2019). Then we have various statistics downloaded from FAOSTAT and World Bank WDI. | Sources of the data are provided in the paper |
| In line 33 we read "One method to partially address spatial mismatch between administrative and other geographic units such as natural hazards". As a matter of fact administrative boundaries are a way to measures land and territoires so there is no mismatch. the Auhtor should rephrase the sentence

in line 90 and further we derive the agricultural GDP as multiplying derived quantity for wholesale price in FAOSTAT: however we don't know if ALL THIS PRODUCTION will be sold, so I would suggest terms as "possible or potential GDP" more than GDP.

  – **RC1, 28 Nov 2022** | Thank you for your comments.

We have modified the text to avoid confusion in the two places where we use "mismatch" in this context.

First, in previous line 15 \| current line 16:
**"Furthermore, a geographic unit of interest, such as the natural area of a river basin, may not align with political administrative boundaries..."**

Second, in previous line 32 \| current line 35:
"**One method to address the case where administrative boundaries and geographic areas of interest are not aligned is to use the gridded (raster) data format.**"

The statistics data we use for disaggregation are national agricultural GDP from the World Bank World Development Indicators (WDI). The WDI data does have limitations and may not include natural losses or self-consumption. We have added text and a link in the footnote for more details about the dataset.

We added text in the third paragraph of Section 2.2, AgGDP Statistics and Linked Grids.
In previous line 195\| current line 225:
**"The World Bank compiles these national accounts data following the International Standard Industrial Classification (ISIC) divisions 1-3 that includes** | We modified text |

**agriculture, forestry and fishing. Given the challenges of compiling national accounts data across the world, limitations include the exclusion of unreported economic activity in the informal or secondary economy. In particular, agricultural output in developing countries may not be reported due to issues such as, natural losses, self-consumption or not exchanged for money. Despite best efforts, agricultural production may be estimated indirectly leading to approximations that are different than the true values. \footnote{See \href{https://data.worldbank.org/indicator/NV.AGR.TOTL.ZS}{World Bank WDI} for more details on metadata and limitations}**"

In previous line 90-183 | current line 96-211, we construct the priors for different components to be used as input for disaggregation of WDI national AgGDP statistics. We have modified the text to emphasize the data construction is a prior in the model and figures.

In previous line 91 | current line 97:
"The **prior for** crop component in the gridded AgGDP is generated by multiplying the quantity of production…"

In previous line 113 | current line 124:
"We calculate **the prior for** the component of livestock production in gridded AgGDP based on…"

In previous line 143 | current line 160:
The value of wood products **prior** per pixel

Figure 1 legend
"Crop production value **prior**"

Figure 2 legend
"Livestock production value **prior**"

Figure 3 legend
"Wood forest production value **prior**"

Figure 4 legend
"Fishery production value **prior**"

| | | |
|---|---|---|
| The Estimating Local Agricultural GDP across | Thank you for your comments. | We modified text on |

| | | |
|---|---|---|
| theWorld paper s a very interesting article, and faces in an innovative way the issue of integrating offical economic statistics, often scarce, with geospatial data available and of high quality.

However, is it suggested a re-wording in the title and in the text for the term "agricultural GDP". Techically speacking Agricultural Value Added (which is a percentage of GDP) is more precise.

Moreover the Authors should better explain how of the total livestock, crops etc. that could teoretically contribute to the Agricultural value added are netted out of the quantity related to natural losses, self-consumption by farmers, or simply are unsold in the market. In all these cases we have a physical quantity that do not reach the buyer, and therefore can't contribute the the agricultural value added as meant by the SNA and economic statistics. If these aspects are not considered by authors, it is suggested to use the term " potential value added".
with these changes and/or further explanations, the article is welcome to be published.
   – **RC2, 06 Dec 2022** | Our understanding is that GDP of a sector is the sum of all value added in the sector. The comment is duly noted, we have clarified in the text and added further explanation in the text to provide a common understanding of the World Bank's definition of agricultural Value Added GDP, including a link to the dataset and metadata.

In previous line 62 \| current line 67:
"**In this paper, we present a high resolution gridded Agricultural GDP (corresponding to "agriculture, forestry, and fishing, value added" in World Development Indicators, henceforth AgGDP)...**"

In previous line 193\| current line 222:
"The national totals are obtained from the **publicly available** World Development Indicators (World Bank, 2019) and averaged over three years around 2010."

We added text in the third paragraph of Section 2.2, AgGDP Statistics and Linked Grids.
In previous line 195\| current line 225:
**The World Bank compiles these national accounts data following the International Standard Industrial Classification (ISIC) divisions 1-3 that includes agriculture, forestry and fishing. Given the challenges of compiling national accounts data across the world, limitations include the exclusion of unreported economic activity in the informal or secondary economy. In particular, agricultural output in developing countries may not be reported due to issues such as, natural losses, self-consumption or not exchanged for money. Despite best efforts, agricultural production may be estimated indirectly leading to approximations that are different than the true values. \footnote{See \href{https://data.worldbank.org/indicator/NV.AGR.TOTL.ZS}{World Bank WDI} for more details on metadata and limitations}** | definition and added a link to the metadata. |
| **1.1 Are the data and methods presented new?**
The AgGDP dataset is unique. The methods used to generate the dataset are not new per se, but it is the combination of | Thank you | |

| | | |
|---|---|---|
| diverse input data and the use of multiple methods to generate AgGDP that is new. I commend the authors for taking on the challenge of producing this important dataset
-RC3 | | |
| **1.2 Is there any potential of the data being useful in the future?**
Yes I am sure the layer will be used in many future global modelling exercises that require detailed agricultural economic activity/output information. See comments in the data quality section that would aid future users in their understanding and usage of the dataset.
-RC3 | Thank you. | |
| **1.3 Are methods and materials described in sufficient detail?**
The methods and data section is comprehensive. I recommend some discussion on the impact of choices/assumptions such as those made on line 144. This is just one example, other assumptions should also be addressed in the discussion. | Line 144 refers to the use of MODIS Land Cover map (Friedl et al., 2010) for year 2011 is overlaid on top of that for year 2010 to detect the area that has changed from forest to non-forest. We added to the Footnote 5 text in **bold**:
"The measurement is limited to detection of land cover change from satellite **at a spatial resolution** and will likely **not** account for selective harvesting or forest degradation. **And the area of forest is considered homogeneous of equal production value. Also, it could result in upward bias when trees are cut down for plantation replanting and not used in further processing of timber production."**

We also have added discussion on the impact of choices/assumptions in the following places:

In previous line 115 \| current line 126:
"**Due to data limitations, distribution maps for other animals such as ducks, horses, camels, and bees are not available. But the FAOSTAT's livestock production values include a more comprehensive list of animals and their products. By distributing FAOSTAT values to grids in proportion to the five** | We modified text |

| | primary livestock species, we assume that other animals included in FAOSTAT have a similar spatial distribution to the five primary livestock species. **This assumption is generally valid, but may not be accurate in special areas, such as deserts where camels are an important source of livestock products.**"

 In previous line 164 \| current line 192:
 "**This is a simplified assumption and may cause overestimation in places where there are inland waterbodies, but not much fishery activities going on.**"

 In previous line 204\| current line 240:
 "**In actuality, the priors that we have constructed do not encompass all elements of AgGDP, and the national and sub-national AgGDP statistics include a broader range of production values. But the priors account for most variation between pixels, and thus their shares can serve as appropriate proxies in the AgGDP disaggregation model.**" | |
|---|---|---|
| **1.4 Are any references/citations to other data sets or articles missing or inappropriate?**
 I did not miss anything. | Ok. | |
| **1.5 Is the article itself appropriate to support the publication of a data set?**
 Yes with modification following the recommendations below. | Ok. | |
| **2.1 Is the data set accessible via the given identifier?**
 The data is accessible at the following location https://datacatalog.worldbank.org/search/dataset/0061507 | Ok | |
| **2.2 Is the data set complete?**
 The data and metadata are incomplete
     −     The metadata is quite sparse (perhaps limited by the WB Data Catalog format) | We do use the WB Data Catalogue format. We have updated the metadata to reflect your suggestions.

     1. We use the metadata from the WB Data Catalog format

     2. We added units | We modified metadata catalogue and submitted the updated records accordingly |

| | | |
|---|---|---|
| − The units are not mentioned in the metadata.

− Why is the dataset floating point? This level of precision seems unjustified – is it float because integer formats cannot deal with the large range of values? Even so, reporting GDP in USD to decimal places seems unjustified.

− The metadata does not link back to the preprint.

− The first published date is after the last updated date – please check and corect.

− Sea areas where no AgGDP data is possible (because marine-based AgGDP is allocated to land) and territories where there is no data available (due to a lack of data for now) are treated the same – this is not very elegant. Consider using different pixels values to distinguish these two "no data" types. | 3. We adjusted to the unsigned integer format "INT4U" as the maximum value of the AgGDP dataset is lower than the 4 294 967 296

4. We added a related link in the WB Data Catalog to the preprint, which was not available when the data record was posted (prior to submission)

5. We will check the difference in posted and last modified date. The updated date is now 02/03/2023.

**6.** We appreciate your suggestion on the differences between No data and marine allocation on land. We modified the metadata to reflect this distinction :"No data" in the case of lack of input data and "Not available" for sea areas (e.g. exclusive economic zone (EEZ)). When the data are available, we can consider collecting detailed information on sea areas for a future version. **The value of marine fisheries production is based on its proximity to fish landing ports weighted by a composite indicator of equal weight from the number of visits and sum of the vessel hold of fishing vessels. So, pixels in sea areas and EEZ are "Not available" and pixels associated with land can have "No data" in the case of lack of input data.** | including the dataset. The update may take some time for posting |
| **2.3 Are error estimates and sources of errors given (and discussed in the article)?**
No. There are no error estimates or validation data in the dataset, though they are discussed in the article. See below for comments on | We have elaborated on the Section **3.3 Validation (current line 375-423)**. A true validation of the predictive accuracy across the world is not possible since we don't have high resolution estimates of AgGDP from sources other than this study. And the flexibility of this model enables us to incorporate all available AgGDP statistics at various levels to | We added text |

| | | |
|---|---|---|
| validation and sensitivitiy analysis. | improve results. But statistics used as an input in the model cannot be used to validate the results anymore.

Nevertheless, we are able to design an experiment and validate the method in Brazil where multiple geographic levels of AgGDP statistics exist. We have added discussion in detail in **the second paragraph in 3.3 Validation (current line 382-395).**

Additionally, we conducted a global comparison of our model and a rural population-based model in **the third paragraph in 3.3 Validation (current line 396-423).** It shows that our model provides more information than the rural per capita model at a granular spatial level. | |
| **2.4 Are the accuracy, calibration, processing, etc. state of the art?**
The pre-processing steps to generate the AgGDP dataset are appropriate. The section on uncertainties in each input layer and how those uncertainties may be compounded when they are combined is rather brief. Some quantification via a sensitivity analysis would be a welcome addition to the paper. The lack of a quantitative uncertainty assessment is a weakness of the paper and the dataset in the absence of a robust validation. | In the original preprint, we had 3.2.1 Regional accounts and 3.2.2 Components to discuss the uncertainty in collected statistics and various input layers. We modified the structure of the paper and related text to be more explicit about the results, uncertainty and validation. Now we changed the title of Section 3 to **3 Results, Uncertainty, and Validation (current line 281-423)**. Its subsection **3.2 Fitness-for-use and uncertainty (current line 314-374)** discusses uncertainty of **3.2.1 Regional accounts** and **3.2.2 Components**. At various places in the text, we also added discussion on the impact of choices/assumptions for components as discussed in **the response to Comment 1.3**.

In **the second paragraph of 3.2 Fitness-for-use and uncertainty**, we discussed sensitivity analysis in related previous work.

In current line 326:
"**In previous work, our team conducted sensitivity analyses and examined consequences of methodological-data choices involved in a cross-entropy model to disaggregate crop production statistics \citep{joglekar_pixelating_2019}. These analyses included eight scenarios that varied in allocation methods, data grouping, input variables, and different levels of statistics. The analysis** | We modified the text and added a map |

| | indicated that allocation results are most **dependent on the degree of disaggregation and quality of the underlying national and subnational production statistics. Therefore, we provide more discussion in section 3.2.1 Regional accounts. Additionally, the results are moderately sensitive to allocation methods. We previously compared three models for the case of Brazil \citep{thomas2019generating} and found that cross-entropy is the most appropriate method for the global study with relatively high accuracy and flexible data requirements, when compared with either the spatial regression or rural population methods. Interested readers may find more details in the Brazil paper. Lastly, the results are somewhat sensitive to the grouping and format of input components that serve as priors, which we discuss in 3.2.2 Components.** "

 Additionally, following Robinson et al (2014) and Yu et al (2020), we have added a map of the average spatial resolution (ASR) in **Figure 6** for subjective uncertainty measure. The average spatial resolution per country is the square root of the land divided by the number of regions with regional account statistics, which is the biggest source of uncertainty in sensitivity analysis as mentioned before. | |
|---|---|---|
| **2.5 Are common standards used for comparison?**
 Th AgGDP dataset is correlated with night time lights (section 2.5). This somewhat contradicts the introduction that states that night timelights are not always a good indicator of agricultural economic activity. The choice is not well justified. I understand that the AgGDP dataset is unique and depends on many input datasets on production value (limiting the availability of possible datasets against which to validate), but I would like to see a much better | As indicated **in the response to the comment 2.3**, we have elaborated on the validation part in **3.3 Validation (current line 375-423)**. We compared different methods and validated the existing approach in the Brazil case (see Thomas et al., 2019). The availability of multiple geographic levels of AgGDP in Brazil enables us to do so. In the Brazil case study, we allocated AgGDP statistics at the microregion level to 5 arc minute grids and then aggregate them to the municipio level (one level below the microregion) and compared with the true data at the municipio level. The correlation between the predicitons and acutal values for the 5564 municipios was 0.91 for our cross-entropy model. We cannot do the comparison globally because many countries lack AgGDP statistics at the subnational level, which is a data requirement for | We modified the text |

choice of validation/comparison with a strong justification too.
The statements on lines 263 onwards are not sufficient indicators of quality. This section can be substantiated by reference to national studies that have also spatially decomposed AgGDP or similar measures of agricultural economic output.
The correlation table (Table 2) is a very high-level aggregation, which is does not reflect the highly spatially disaggregated AgGDP and Night Lights data. It is not very convincing or useful. The validation section seems to compare this cross-entropy model against spatial allocation model based on rural population. The description of the rural population based comparison dataset is not sufficient for a reader to fully understand what it is, how it was made and thus what is being compared against what. I assume that national and subnational Ag GDP is disaggregated based on head count giving every rural person an equal share of the AgGDP? But I am guessing.
Either way, the validation is a case of comparing one model against another with the argument that the assumptions in one model are more valid than those in another. This is not a very satisfactory validation and I amnot sure what message is intended by showing the two have different degrees of correlation in differnet parts of the world.

the analysis. Therefore, at the global level, we only compare the model to a rural population-based model and illustrate that our model provides more information than the rural per capita model. We have added more text to explain the rural per capital model in **the last paragraph of 3.3 Validation (current line 396-423).**

We compared our estimates to nighttime lights and rural population to demonstrate that the latter two are not good proxies in low-income regions and at lower administrative levels, e.g., Sub-Saharan Africa and administrative geographic level 2, where AgGDP data are not readily available before our work. Our estimates provide more information. As shown in the Brazil case, the cross-entropy model performs better than the rural per capita model at lower administrative levels.

Previous lines 263 onwards are a general description of the results. We don't intend to use them for official validation and have made it clearer now in the revised version.

We focus on AgGDP areas with values above 200,000, excluding the Low Agricultural GDP/NA category where the measurement of rural population and AgGDP may have discontinuity due to modeling inaccuracies. In current line 416:
**"We use a Spearman correlation for a 3 x 3 window of pixels with a focus on AgGDP areas with values above 200,000, excluding the Low Agricultural GDP/NA category where the measurement of rural population and AgGDP may have discontinuity due to modeling inaccuracies."**

We have provided regional statistics on the correlation between our cross-entropy results and nighttime lights and rural population in **Table 2**. We can provide data of rural per capita model and the correlation map based on request. But as illustrated in the Brazil case, the cross-entropy results are better. So our priority is to provide the cross-entropy results and its related input data.

| | | |
|---|---|---|
| Why exclude areas from the analysis with values that are less than 200,000 (USD?) ?. No summary or regional statistics on the correlation are provided. This cprrelation map and the rural per capita GDP should be provided as spatial datasets with the AgGDP dataset. | | |
| **2.6 Is the data set significant – unique, useful, and complete?** From my perspective the AgGDP dataset is unique. I have given recommendations above to make it both useful and complete. In addition to that, I recommend that a table of the production values (priors) per country and the collated GDP data would be very valuable additions to the dataset. Where appropriate these layers should also be provided in spatial data formats. This would help users understand the spatial patterns and artifacts in the AgGDP (alloc.tif) dataset and help ensure appropriate use | Thank you for your advice. We have provided a table of the share of priors for countries with measurable AgGDP in the **Appendix Table B8** as mentioned in current line 320. We use the naming convention and current boundaries of the World Bank that excludes disputed areas and a few small islands from the table.  We also provided the different components of priors in raster format in the WB Data Catalog, which will allow users to better contextualize and ensure the fitness-for-use of their application(s). The maps are production values; however, we also include a Table (B8) of the share of component priors, which facilitates comparison across the components. | We added Table B8 |
| **3.1 Are there any inconsistencies within these, implausible assertions or data, or noticeable problems which would suggest the data are erroneous (or worse). If possible, apply tests (e.g. statistics). Unusual formats or other circumstances which impede such tests in your discipline may raise suspicion.** I note the edge effects above which could give potential users pause for thought before using this data The authors could subnational representations of the data, both spatial and tabular which | Point well taken. We provide summary tabular data at the administrative level, which will increase awareness of the results and provide another format for increased use. It will be available at the World Bank Data Catalog. Data in tabular format at admin 1 level. We will use current publicly available World Bank boundaries to be consistent with WB Data Catalogue requirements. The GAUL data require a license.  If the edge effects refer to the clear-cutting boundaries in some places, such as Western Sahara, The Northwest Territories of Canada, and a few subnational divisions, that's a result of the model mechanisms and input data limitations. The model takes in a subnational division or a country as a unit and allocates the statistics to its land. These clear-cutting boundaries are the administrative | The WB Data Catalog is updated accordingly |

| | | |
|---|---|---|
| would make it easier to assess whether there are any noticeable problems due to modelling or assumptions. | boundaries and they become very distinct when neighboring units are very different. For example, AgGDP statistics for Western Sahara are not readily available, whose neighboring countries have data, so we can see that it is cookie cut from the map. Also, one of the input layers, the gridded livestock (Robinson et al., 2014; Gilbert et al., 2018), shows distinct administrative boundaries lines at some places. Since this information serves as part of priors, it will bring this effect to our results as well. In this case, the edge effects exist with small values. Major variations are not affected. We added a note to explain the edge effects on the data download webpage.

"**In some cases, the data may illustrate the edge effects referring to the clear-cutting boundaries in some places and a few subnational divisions, which is the result of the model mechanisms and input data limitations. The model takes in a subnational division or a country as a unit and allocates the statistics to its land. These clear-cutting boundaries are the administrative boundaries and they become very distinct when neighboring units are very different.**" | |
| **3.2 Is the data set itself of high quality?**
The effort is impressive; detailed estimates of Ag GDP are valuable.
The datasets value is detracted from by the lack of validation data, spatial artifacts and the presentation of the data (level of precision is not justified, file name choice, sparse metadata, lack of tabular summaries). | Thank you. Following your advice, we have strengthened the validation section, explained the spatial artifacts, and improved the data presentation, as explained in previous responses. | We have modified the text |
| **4.1 Is the data set usable in its current format and size?**
The format is suitable for use in both open and proprietary GIS software and can be easily read in open source sofware such as R or Python for statistical analysis | Thank you. We have explained the edge effects in **the response to Comment 3.1**. This is the result of the input data (lack of AgGDP in some places and edge effects in livestock data). We added an explanation to the metadata.

We changed the filename to "**aggdp2010.tif**" and change the dataset to integer "INT4U" format. We | We have modified and added information and files in the World Bank Data Catalog |

| | | |
|---|---|---|
| There are edge effects in northern latitudes and on some country and subnational boundaries that are rather inelegant – can these be dealt with better?
Recommend to change the file name from alloc.tif to something more meaningful
Recommend the dataset is converted to integer not float – both to reduce size and to be more realistic about the precision og the GDP estimates. Estimates could even be rounded up to the nearest 1000 USD.
Recommend that the different types of no data are treated differently | uploaded the following information and files to the World Bank Data Catalog.

"**The Global Gridded Agricultural Gross Domestic Product (AgGDP) datasets provide information on agricultural GDP across the world. The global gridded AgGDP 2010 dataset at approximately 10x10 km is the result of a data fusion method based on cross-entropy optimization. We disaggregate national and subnational administrative statistics of Agricultural GDP (2010) into the global gridded dataset at using satellite-derived indicators of the components that make up agricultural GDP, namely crop, livestock, fishery, hunting and timber production. The data resources include the gridded global estimates at approximately 10x10 km and the priors that we derived in the model: crop production value as a prior, livestock production value as a prior, forest production value as a prior and fish production value as a prior. Data resources includes: Global gridded AgGDP (ggdp2010.tif), Global gridded crop production value as a prior in the model (aggdp2010_crop_prior.tif), Global gridded livestock production value as a prior in the model (aggdp2010_ls_prior.tif), Global gridded forest production value as a prior in the model (aggdp2010_forest_prior.tif), and Global gridded fish production value as a prior in the model (aggdp2010_fish_prior.tif)** | |
| **4.2 Are the formal metadata appropriate?**
See previous comments on metadata completeness – this needs to be addressed. | Thank you. We have addressed these comments. | We have modified text |
| **5.1 is the length of the article appropriate?**
Length is fine, but more space can be given to (i) a more robust validation or (ii) a sensitivity analysis to understand the impact of choices in methodology and/or the contribution of the | Thank you. We have revised section **3 Results, Uncertainty, and Validation (current line 281-423)**.

**3.2 Fitness-for-use and uncertainty** discusses sensitivity analysis of methodological and data choices in our previous examination of the cross-entropy model. We also elaborate on uncertainty from three biggest sources, including regional statistics of AgGDP, allocation models, and various component priors. | We have modified text |

| | | |
|---|---|---|
| uncertainties in the input layers. | **3.3 Validation** presents our validation based on a country study in Brazil and a global study against the rural per capita modelling results. The Brazil case study validates our methodology and the global study against the rural per capita results validates that our global results are in a reasonable range. | |
| **5.2 Is the overall structure of the article well structured and clear?** Structure is fine though the natural hazards component seems like an add-on that does not add much value to the paper and dataset, which is really about AgGDP. The hazards part is one of many possible applications. Is it essential to the paper to focus on one use case like this? If a use case is a requirement of the journal then fair enough. Starting the conclusions section with a paragraph on hazards is a curious choice given that this is not the core purpose of the paper. Again if this is a requirement of the journal then fair enough. | We have restructured the paper to focus on AgGDP dataset and reduced the text on the natural hazard component, where we use it only as an illustration of data usage in the section **4 Illustration of use: drought risk and water scarcity**. Accordingly, we have modified the section **5 Conclusion** to focus on the data and only present the hazard case as one of the many possible applications. | We modified the text |
| **5.3 Is the language consistent and precise?** The language would benefit from professional English editing. The text is largely understandable but many lines in the text jar due to non-standard English. This reduces the readability. I had to pause and re-read some lines several times, e.g., lines 17 and 18. The dataset and documentation is an extremely valuable resource and I commend the author's efforst for developing it; please bring the text up to the same level of value as the data. Line 8 | We have reviewed and revised the text to improve its readability. Among the changes, the following are listed in response to this comment: In previous line 17 \| current line 19: "**Around five billion hectares of land is dedicated to agriculture, but collecting and reporting data in areas affected by fragility, conflict, and violence can be challenging, resulting in incomplete or outdated geographic coverage.**" In previous line 8 \| current line 8: "**To illustrate the use of the new dataset, the paper estimates the exposure of areas with at least one extreme drought during 2000 to 2009 to agricultural GDP, which amounts to around US\\$432 billions of agricultural GDP circa 2010, with nearly 1.2 billion people living in those areas.**" | We modified the text |

The paper estimates the exposure of areas with at least one extreme drought during 2000 to 2009 to agricultural GDP is an estimated US$432 billion of agricultural GDP circa 2010, where nearly 1.2 billion people live.
Alternative We estimate that US$432 billion of agricultural GDP (circa 2010) was exposed to at least one extreme drought during 2000-9.

If hazard exposure is important, consider adding it to the title.
Line 2 of the abstract is hard to parse
Line 6 – consistency needed – either small "a" on agricultural GDP throughout the paper or capital A.
Line 12 remove "the" , same in line 15, same in line 92 and many other instances of non-standard use of the definite article Check and correct throughout the text.
Line 15 – location variation in what?
Line 15 – the possible implications of the mismatch are not clear
Line 72 –AgGDP not agricultural GDP – check paper that this abbreviation is used henceforth.
Line 79 – what does efforts varied mean?
Line 93/94 is the repetition necessary? Aim to be concise.
Line 118 – clarify the pixel areas. Is this land area or simply the total area of each 5 min pixel? Depends on how the densities were computed, but this is not clear from the paper.

In previous line 2 | current line 2:
"**However, these measures may lack sufficient local variation for effective analysis of local economic development patterns and disaster exposure to natural hazards.**"

In previous line 6 | current line 6:
We changed "Agricultural GDP" to "**agricultural GDP**".

In previous line 12 | current line 13:
We think "the" before "Food and Agriculture Organization of the United Nations" is necessary because it refers to a specific organization. The "the" before "agricultural sector" also indicates that the text is referring to a specific sector.

In previous line 15 | current line 15:
We removed "the" in "the local variation" and added "in production activities". Now it reads "**Yet, economic statistics of the agricultural sector are frequently produced at a national or lower administrative level and may not adequately capture local variation in production activities.**"

In previous line 92 | current line 98:
We removed the "the" before "producer prices".

In previous line 15 | current line 16:
"**Furthermore, a geographic unit of interest, such as the natural area of a river basin, may not align with political administrative boundaries, limiting the ability to conduct a comprehensive overlay analysis of the area.**"

In previous line 72 | current line 76:
Changed to "**AgGDP**". Also checked the rest of the paper and made it consistent.

In previous line 79 | current line 85:
We revised the text for clarity. "**Given the limited availability of data and the global scope of the study, we made various efforts to adjust official statistics and create priors for different components based on the available data.**" Various efforts refer to the fact that for some countries we may have data in the right format, while for other

| | | |
|---|---|---|
| Line 125 – first sentence seems superfluous. Also was the start of civilisation really the first use of forest resources? What about hunter/gatherer societies?
Some statements are superfluous and can be removed. For example
Line 270 The correlation of AgGDP with night light varies across world regions as it requires areas to emit light (Table 2).
Line 279 The exposure to drought is not uniform across the world.
Lines 303-304 are more or less repeated in lines 317-318. | countries we may need to adjust the currency, the year, and align the administrative units to our shapefiles. Also, it refers to different approaches we used on calculating different components.

In previous line 93 \| current line 103:
The only repetition is "SPAM is a cross-entropy model". We think it is necessary because we mention SPAM cross-entropy model in Line 66/67 for explaining our AgGDP estimation method, while in Line 93/94 we elaborate on the SPAM model to provide necessary information for our crop component input. It's important for the readers to know how the input is produced in order to decide proper usage and limitations of our results.

In previous line 118 \| current line 134:
It is the total area of each 5 min pixel. We have clarified in the text.

In previous line 125 \| current line 143:
We revised the text as "**People have utilized forest resources for a long time throughout history for their livelihood and various other purposes.**" We think it's good to highlight the long history that human have been relying on forest resources.

In previous line 270 \| current line 300:
We revised the text as "**We find that the correlation of AgGDP with night light varies across world regions, with Sub-Saharan Africa and the Other Region showing lower correlation values (Table 2).**"

In previous line 279 \| current line 440:
We think this sentence is necessary because it summarizes our findings and starts the discussion in the paragraph.

In previous line 303-304:
We have revised the validation section, and the repetition has been deleted. | |
| **5.4 Are mathematical formulae, symbols, abbreviations, and units correctly defined and used?**
Yes | Ok. | |

| | | |
|---|---|---|
| **5.5 Are figures and tables correct and of high quality?** Maps are clear. Figure captions do not need to start with "This map…" Just state what the figure shows. Recommend an Equal Area projection and remove the E and N coordinates and graticules– they do not add useful information. Check capital letter usage in map legend title; production instead of Production Table 1 is shown before it is referenced in the text. Check capital letter usage in column headings in tables. Table 1 caption is not self explanatory – conversion factor should be explained. | We have modified the map caption and reproduced the figures in R script dropping coordinates and graticules. We also reproduced the maps in an equal area projection (Eckert IV) for the presentation of the maps. We added text to footnote 13 in the results and validation section: **"For presentation in the paper, the coordinate system of the maps is Eckert IV and transformed in R software."** We have used "production" instead of "Production" in map legend titles. We have moved Table 1 after the text referencing the table. We have used capital letters in column headings in tables. We have added explanation for conversion factors in current line 133 and footnote 3: "**The conversion factors reflect biomass differences between different animals \footnote{The uniform conversion factors may oversimplify local variations in livestock patterns. Future work may consider using country-specific values of livestock products from FAOSTAT.}.**" | We modified the maps and captions |
| **Rating** On a scale of 1 (excellent) to 4 (poor) I would give the datasets and paper a 2.5 at the moment with the potential to be closer to 1 than 2. The dataset is **unique**, **potentially significant** and will be **widely used**. To reach it's full potential users need to fully grasp how it was made, what the inputs were, where the major uncertainties are and thus how to properly use the dataset in further research. These are areas for improvement (**completeness and data quality**) in the manuscript and associated | Thank you very much for your thorough comments and constructive feedback. | |

| | | |
|---|---|---|
| datasets that could be included with the AgGDP layer. The **presentation quality** of the manuscript can be improved - see comments above to do justice to the impressive work conducted so far to produce this unique global spatial dataset. | | |
| There clearly is a great effort behind this manuscript, but I feel the authors should focus on the refinement of the methodology in view of producing something that can be more easily updated to account for the dynamicity of the agricultural sector. The authors generate a spatial distribution of the agricultural GDP circa 2010 but this information can hardly be useful for analysis of the risks that the agricultural production face more than a decade later given that the authors themselves highlight the dynamicity of the sector. Besides, the analysis of the exposure to drought falls short to describe the adaptive capacity that characterize many agricultural systems. The authors indicate that the results of the analysis are suitable for global, continental and regional analysis but not for local analyses and one may wonder if in the end this effort is worth doing, also considering that this information cannot be easily validated and that is not suitable to inform local planning. There are some methodological issues that would require in my view some attention: for instance the | Thank you for your comments. We understand it would have been great if our data product was more updated. But when we started the project in 2017, appropriate and available data centered around 2010. However, as data availability has improved rapidly in recent years, we believe it would be relatively easy to update this product to a more recent year when funding resources are available. As like other global gridded datasets, we caution users if they are going to use the data at the local scale. Our dataset aims to capture general variation at local scale, but may miss special local conditions and have local discrepancies. But we believe having a globally consistent and relatively high resolution AgGDP data is a useful base to facilitate related research and policymaking.

We have addressed your recommendations as follows:
      1.     We have revised the section **3 Results, Uncertainty, and Validation**. **3.2 Fitness-for-use and uncertainty** discusses sensitivity analysis of methodological and data choices in our previous examination of the cross-entropy model. We also elaborate on uncertainty from three biggest sources, including regional statistics of AgGDP, allocation models, and various component priors. **3.3 Validation** presents our validation based on a country study in Brazil and a global study against the rural per capita modelling results. The Brazil case study validates our methodology and the global study against the rural per capita | We have modified text |

spatial analysis of the agricultural GDP that is done for the wood products; collinearity of input data. The paper does not contain a quantification of the uncertainties and does not mention the fact that agricultural GDP cannot capture well agricultural production in the informal or secondary economy. A more consolidated discussion of the limitations of this product would greatly benefit the strength of this paper.

Overall, I would welcome the publication after a revision that addresses these main points: 1) Strengthen the methodology or justify better some of the technical choices that were made; 2) Provide some quantification of the uncertainties. 3) Add some discussion on how the ever-growing release of new and better inputs data (crop maps; livestock distribution; dynamic land cover maps; more spatially-disaggregate and recent statistics) may be integrated into this product to reduce or even better to keep up with the temporal mismatch in agricultural production. Finally, one suggestion: the linkages between drought and agricultural production are less important for fisheries and wood production than for crop and livestock whereas the water crowing index is more associated with the distribution of the population, which is itself quite outdated in this analysis. My suggestion would be to remove or shorten the

results validates that our global results are in a reasonable range.

2.      As mentioned in the revised manuscript, in previous sensitivity analysis of the cross-entropy model we found that the biggest uncertainty comes from the resolution and quality of statistics. Thus, following Robinson et al (2014) and Yu et al (2020), we have added a map of the average spatial resolution (ASR) in **Figure 6** for subjective uncertainty measure. The average spatial resolution per country is the square root of the land divided by the number of regions with regional account statistics.

3.      We have added a discussion on future work that may use new and better inputs data for AgGDP statistics and various components **in the last paragraph of the paper**.

**4.**      We have shortened the discussion on exposure to drought and water scarcity to just highlight examples of potential data applications and the suggested linkages in the current **Section 4**.   **These two indicators provide an illustrative example of different linkages to agricultural production. Drought highlights the linkages to crops and livestock whereas water scarcity focuses attention on the distribution of population.**

5.      We have added discussion on the limitations that agricultural GDP cannot capture well agricultural production in the informal or secondary economy, as stated in the current line 225-230.  "**The World Bank compiles these national accounts data following the International Standard Industrial Classification (ISIC) divisions 1-3 that includes agriculture, forestry and fishing. Given the challenges of compiling national accounts data across the world, limitations include the exclusion of unreported economic activity in the informal or secondary economy. In particular, agricultural output in developing countries**

| discussion on the exposure to drought. I understand that it was used as an example of application but in my opinion doesn't really bring much value to the discussion. | **may not be reported due to issues such as, natural losses, self-consumption or not exchanged for money. Despite best efforts, agricultural production may be estimated indirectly leading to approximations that are different than the true values. \footnote{See \href{https://data.worldbank.org/indicator/NV.AGR.TOTL.ZS}{World Bank WDI} for more details on metadata and limitations}**" | |

Referee #3

| This paper extends the authors' previous Brazil-focused work on gridded agricultural GDP to the entire world. It uses cross-entropy optimization to disaggregate an impressively large number of national and subnational datasets to 5 arc-minute grid cells. The authors previously applied this method to the development of gridded global crop data ("MapSPAM"). Their development of gridded agricultural GDP in this paper is a valuable advance given that agriculture is essential to human survival, remains the dominant economic sector in rural areas of most countries (especially low-income countries), and is threatened by climate change. The authors necessarily make some strong assumptions to construct this new dataset, but I don't view their assumptions as being any less tenable than those that underlie standard national accounts statistics. | Thank you for your comments. And we have added text to provide more detail on the limitations of national accounts statistics in **the third paragraph of 2.2 AgGDP Statistics (current line 225-230) and Linked Grids** and **3.2.1 Regional accounts (current line 338-357)**. | We modified the text |
| My overall reaction to the paper is favorable, but I have two general suggestions for making it easier to understand and more convincing. First, it can be organized better. It flows naturally through section 2.3. After that point, I suggest reorganizing it as follows:
 − Create a new section 3, titled "Results and Validation." This section would begin with the presentation and interpretation of the new dataset on gridded agricultural GDP, which is | We appreciate the suggestion on improving the readability of the paper. We have modified the structure of the paper following your suggestions. For section 3, we even add a subsection on uncertainty. Now we have **3 Results, Uncertainty, and Validation** and **4 Illustration of use: drought risk and water scarcity.** | We modified the text |

| | | |
|---|---|---|
| displayed in Fig. 7. It would then compare the new dataset to the night-time lights (NTL) data, the point of which (as I understand it) is to demonstrate that NTL is not a good proxy for gridded agricultural GDP. Nor are gridded total GDP or gridded population (Table 2). Hence, the new dataset does indeed provide new information. The material in current section 2.5 would be integrated into this new section.

    –    Confidence in these findings depends on the validity of the new dataset, so new section 3 would next cover validation. This subsection would begin with the acknowledgment of limitations of the new dataset presented in current section 3.2 (including 3.2.1 and 3.2.2) and wrap up with the presentation of the validation findings in current section 3.1 (including Fig. 10).

    –    A new section 4 would follow and would be titled something like "Illustration of use: drought risk." It would integrate information from sections 2.4 and various parts of section 3 and would include Fig. 5, 6, 8, and 9 and Table 2.

    –    The paper would finish with the existing concluding section. | We have modified **Table 2** to highlight that the new dataset does provide new information. Correlations are notable at the country level, however diminish at the administrative 2 level. The relevant text is in line 304: **"However, notable differences exist between geographic levels. The mean correlation of AgGDP with night time lights (NTL) and population (pop) derived from administrative level 2 data is lower than the national level, which presents evidence of new information from the AgGDP dataset."**

In addition, we provide new text in line 307 on the limitations of the NTL and population proxies: **"Furthermore, limitations exist with these commonly used datasets for applications of AgGDP. For night time lights, Li et al. (2020) provide a cautionary note about rural applications where the presence of agricultural activities typically takes place. A population model assumes proportional activity to population by strata (e.g. rural), which does not account for the type of rural of agricultural activity, and the model requires a standard definition of rural, which can pose challenges in global applications (e.g. stylized facts in the urban and development economics literature Roberts et al., 2017). Notably, the rural population dataset also has variation in the geographic level of the input information and currency across the world, especially when dependent on the availability of a population census. Also, the AgGDP dataset may attenuate modeling concerns of endogeneity when using AgGDP along with population or night time lights."** | |
| Second, and more substantively, the authors need to address several issues with the construction of the wood production component in section 2.1.3: | | |

| | | |
|---|---|---|
| – The Lebedys and Li (2014) estimates used by the authors are, to my knowledge, the best available estimates of forest sector GDP, but they focus on industrial roundwood (and products derived therefrom) and largely exclude fuelwood, which accounts for half of global wood harvests. As a result, even allowing for fuelwood's unit value being much lower than industrial roundwood's, the current wood production component underestimates the contribution of wood harvests to agricultural GDP. The easier option for the authors would be to stick with the current estimated component but acknowledge that it underestimates the wood harvest value. The harder option, but the one I encourage the authors to consider, is to figure out a way to add the value of fuelwood harvest to the component. Fuelwood harvests are usually correlated with the collection of nonwood forest products, so perhaps the authors can use information in Siikamaki et al. (2015) to impute gridded values for fuelwood harvests. I note that Siikamaki et al. refer to some of the studies they reviewed as having included information on fuelwood values. Annual data on national harvests of fuelwood from FAOSTAT-Forestry might also be useful in the imputation. | Thank you for sharing your expertise in the forestry sector. We have considered your second option, but we don't have reliable data on wood fuel production value by country (FAO has estimates for quantity but not value) and not sure about the correlation coefficient and significance between fuelwood and non-wood forest products, so we will consider it for future work. Following your first option, we have added text to acknowledge the underestimation of wood harvest value in **the last paragraph of 2.1.3 Forestry production and hunting (current line 174-183)**.

"**In our analysis of the forestry sector GDP, we have utilized the estimates provided by Lebedys and Li (2014) as the best available source. However, it should be noted that these estimates primarily capture activities within the formal forestry sector and do not take into account the value-added generated by informal activities such as wood fuel production and non-wood forest products. To account for non-timber forest products, we have utilized the estimates provided by Siikamaki et al. (2015). Despite these efforts, it is acknowledged that the current analysis may still underestimate the forestry sector GDP due to the lack of reliable data on fuel wood production, which could account for half of global wood harvests. This is a common issue as fuel wood values are often not properly captured in official statistics, as they are often collected for subsistence or sold in remote rural areas in many countries (Lebedys and Li, 2014). In future research, we intend to make efforts to acquire more reliable data on fuel wood production to improve the accuracy of our estimates of the forestry sector GDP.**" | We modified the text |

| | | |
|---|---|---|
| – The authors write, "The value of wood products per pixel is calculated based on forest loss from year 2010 to year 2011 …." This statement requires qualification and, ideally, some additional analysis. The MODIS dataset the authors use to calculate "forest loss" measures tree cover, which includes perennial tree crops such as oil palm plantations, cocoa plantations, orchards, etc. in addition to wood-producing forests. This is a well-known deficiency of satellite-based "forest cover" datasets (Tropek et al. 2014; https://www.science.org/doi/10.1126/science.1248753). The authors' estimate of "forest loss" thus includes the replanting of perennial tree crops that occurs when the trees have reached the end of their economic lifetime. The resulting upward bias in "forest loss" can be substantial. For example, oil palm is replanted every 20-30 years, which implies a 3-5%/year "deforestation rate" that is many multiples of the annual loss rate for true forests reported in standard sources (e.g., FAO's Global Forest Resources Assessment). Fig. 3 in the paper illustrates this problem, as it shows wood production occurring in parts of Malaysia and Indonesia that are virtually 100% oil palm plantations. I know there are remote sensing products that show the locations of oil palm plantations, and perhaps there are ones for other non-forest tree crops too. I encourage the authors to use these products to estimate forest loss more accurately by masking out areas with tree cover that are not forests. | Thank you. We have added text in **Footnote 5** to acknowledge the limitations of satellite-based forestry change detection: "The measurement is limited to detection of land cover change from satellite **at a spatial resolution** and will likely **not** account for selective harvesting or forest degradation. **And the area of forest is considered homogeneous of equal production value."** Even though forest loss measured by satellite has limitations (Tropek et al. 2014), it is globally measured and publicly available.

For the case of plantations, we agree that our method may overestimate timber values where plantation trees were cut down for replanting. But sometimes even though trees (oil palm, coffee, orchard, etc.,) in plantations may need to be cut, trunks may still provide timber value. For example, orchard wood from trees such as pear (Fotin and Cismaru, 2011) and walnut (Aletà, 2013) have desirable properties and are popular for furniture production. And oil palm wood can be used for woven, furniture, and building materials (Victoria and Tamang, 2007). Malaysia is using more wood from oil palm and rubber trees for veneer, plywood and panels (Yusof, 2017). It depends on plantation types and local practice on how the trunks are being used. We leave it to future work to more carefully examine trees that are cut down for plantation replanting and not used for further processing in timber production. For now, we just include this caveat in the **Footnote 5:** "**Also, it could result in upward bias when trees are cut down for plantation replanting and not used** | We modified the text |

| | **in further processing of timber production.**" | |
|---|---|---|
| | Aletà, N. (2013, July). Using walnut species for timber production in southern Europe. In *VII International Walnut Symposium 1050* (pp. 383-388). | |
| | Fotin, A., Marthy, M., & Cismaru, I. (2011). Study Concerning The Influence Of Milling Parameters Upon The Surface Quality Of The Birch And Pear Wood. *International Conference of Scientific Paper AFASES*. | |
| | Victoria, T. C., & Tamang, P. (2007). Oil Palm and Other Commercial Tree Plantations, Monocropping: Impacts on Indigenous Peoples' Land Tenure and Resource Management Systems and Livelihoods. In *United Nations Permanent Forum (UNPFII) on Indigenous Issues*. | |
| | Yusof, A. (2017). Use of wood from rubber, oil palm trees increasing, says MTIB. *New Straits Times*. Accessed January 13, 2023 from https://www.nst.com.my/business/2017/12/312606/use-wood-rubber-oil-palm-trees-increasing-says-mtib | |
| − Also requiring qualification is the statement, "forest loss due to fire should be removed because it does not result in wood products." Land clearing often involves a first stage of wood harvests followed by burning to eliminate remaining vegetation and woody debris. The authors' assumption that wood harvests do not occur in areas with fires thus results in underestimating the area harvested for wood products. I can't think | Thank you. We added this caveat in **Footnote 6.**

**"Still, sometimes wood harvests may occur in area with forest fires, and therefore the elimination can underestimate the area harvested for wood products."** | We modified the text |

| | | |
|---|---|---|
| of a way to fix this problem, but the authors should acknowledge it. | | |
| Abstract: State the year of the new gridded dataset, i.e., 2010. Precede the penultimate sentence on the drought analysis with a phrase like, "To illustrate use of the new dataset, the paper …." Such a phrase would clarify that the paper is not primarily about drought risk. The paper would need to be completely rewritten if that were the case. | Thank you. We have added the text "**for the year 2010**" in line 6. Additionally, we have revised the text as **"To illustrate use of the new dataset, the paper estimates…"** | We modified the text |
| Line 22: The authors could note that detailed agricultural data are also needed to evaluate forest restoration opportunities (e.g., P. Shyamsundar et al., "Scaling smallholder tree cover restoration across the tropics," Global Environmental Change 76, 2020; https://doi.org/10.1016/j.gloenvcha.2022.102591), which have become a focus of "nature-based" climate solutions (B. Griscom et al., "Natural climate solutions," Proc Natl Acad Sci USA 114, 2017; https://doi.org/10.1073/pnas.1710465114) since the launch of the UN Decade on Ecosystem Restoration (https://www.decadeonrestoration.org/). Agriculture is the main land-use competitor for forestry. The dataset developed in the paper will help researchers and policymakers better understand the opportunity cost of converting land from agriculture to forest. | Thank you. We have added the text and reference in line 23 in the current version "**evaluation of forest restoration opportunities (Shyamsundar et al., 2022) as part of nature-based climate solutions (Griscom et al., 2017)**". | We modified the text |
| Line 45: Given that the paper is about GDP, "income" would be better than "wealth." | Thank you. We have modified the text in line 48 of the current version following your suggestion. | We modified the text |
| Line 50: The phrase "the uniform distribution of labor in agriculture is another key concern" is vague and should be clarified | Thank you. We have revised and elaborated the text in line 53 of the current version: "Also, **the strong assumption of** uniform distribution of labor in agriculture is another key concern (Gollin et al., 2014). **Uneven agricultural productivity across different regions or locations can lead to a non-uniform distribution of labor within the** | We modified the text |

| | sector, which has implications for the accuracy and effectiveness of models based on rural per capita allocation." | |
|---|---|---|
| Line 65: The authors refer to two main contributions of the paper, with one being the drought analysis. I view that analysis as an illustration of the use and value of the new dataset, not as a main contribution. For the latter to be the case, the authors would need to provide more context for the drought analysis and evaluate it more directly against prior analyses. Constructing the new dataset is a sufficient contribution to justify publication of the paper in my view. | Thank you. We have modified text to one main contribution with an illustration as shown in line 70 of current version. | We modified the text |
| Line 92: State the year of the producer price data. 2010? Mean of 2009-2011? Relatedly, the authors need to explain somewhere whether their agricultural GDP estimates are purchasing power parity (PPP) estimates or market-price estimates. Information in footnote 9 is pertinent to this point. The authors should explain the implications for interpretation of the dataset if the prices that underlie it are not using consistently defined (i.e., some prices are in PPP terms while others are market prices). | For the producer price, we take the average of prices between 2009-2011 as the baseline. But due to missing data for certain countries, crops, and years, sometimes it may be the average of three years, or two years, or one year during 2009-2011, or sometimes the closest year available. We added text in **the first paragraph of 2.1.1 Crop value of production**.

In current line 99-103:
"**As for the producer price, ideally, we need sub-national level figures since prices for agricultural products can vary greatly within countries and their subdivisions, but such a dataset is not available globally. Therefore, we use the FAOSTAT's national producer prices and take the average of 2009-2011, in order to mitigate the potential impact of temporal variation. However, due to missing data for certain countries, crops, and years, this average may be based on a smaller time period or the closest year available.**"

As stated in line 262 in the original manuscript and line 284 in the current version, we estimate the AgGDP value in **constant US$ 2010** for each 5-arcminute grid. And as stated in current line 243, | We modified the text |

| | | |
|---|---|---|
| | different measurement units are all converted to constant US$ 2010 using appropriate deflators and exchange rates before they are used in the final allocation model. | |
| Footnote 3: This point should be incorporated into the text. Prices for agricultural, forestry, and fishery products can vary greatly within countries and their subdivisions. The use of national prices is unavoidable given current data limitations, but it is a shortcoming of the new dataset that the authors should acknowledge in the text. | We modified the text to bring the footnote into **the first paragraph of 2.1.1 Crop value of production**.

In current line 99-103:
"**As for the producer price, ideally, we need sub-national level figures since prices for agricultural products can vary greatly within countries and their subdivisions, but such a dataset is not available globally. Therefore, we use the FAOSTAT's national producer prices and take the average of 2009-2011, in order to mitigate the potential impact of temporal variation. However, due to missing data for certain countries, crops, and years, this average may be based on a smaller time period or the closest year available.**" | We modified the text |
| Line 116: The use of uniform livestock conversion factors across countries seems like an unnecessary simplification. Why not use country-specific FAOSTAT data on the value of products from each type of animal? | Thank you. Following FAO (2011 & 2019), we applied the international Livestock Units (LSU) from Eurostat (2018) to aggregate livestock of various species and facilitate comparison across countries. The LSUs for the five species we included in our study are similar across different regions, so we used uniform conversion factors. Future versions will consider improving the work by using country-specific values of livestock products from FAOSTAT. For this version, we added caveats in **Footnote 3**:

"**The uniform conversion factors may oversimplify local variations in livestock patterns. Future work may consider using country-specific values of livestock products from FAOSTAT.**" | We modified the text |

| | FAO, 2011. Guidelines for the preparation of livestock sector reviews. Animal Production and Health Guidelines No. 5; Food and Agriculture Organization of the United Nations (FAO]: Rome, 2011. Available at: http://www.fao.org/docrep/014/i2294e/i2294e00.pdf.

FAO, 2019. FAOSTAT Agri-Environmental Indicators – Livestock Patterns, http://www.fao.org/faostat/en/#data/EK | |
|---|---|---|
| Lines 130-132: The authors' use of forestry terms is unconventional. I recommend the following rephrasing: "The trees are harvested for fuelwood and industrial roundwood, which is processed into a variety of products including lumber, plywood, furniture, and paper products." Mentioning fuelwood is necessary given that it accounts for half of global wood harvests. | Thank you. We have modified text as recommended in line 148-150 of current version. | We modified the text |
| Footnote 5: The MODIS land cover data used by the authors is quite coarse, ~500 m at the Equator. I doubt it reliably measures selective harvesting or forest degradation. I recommend rephrasing the footnote as follows: "The measurement is limited to detection of land cover change from satellites and might not fully account for selective harvesting or forest degradation." | Thank you. We have modified text as recommended in Footnote 5. | We modified the text |
| Lines 188-189: Mention the typical level of the subdivisions in the dataset here or earlier. Lines 230-231 imply they are mostly Level 1 subdivisions (i.e., states or provinces). | Thank you. We have modified text to include a descriptive of the typical administrative level in line 217 of the current version:

**"The typical administrative level is at the state or provincial level."** | We modified the text |
| Lines 214-219: The authors state, "Theoretically, the sum of these components should be close to the official values obtained from the World Development Indicators." This statement prompts two thoughts. First, as part | We have provided a table of the shares of priors for countries with measurable AgGDP in the **Appendix Table B8**. We use the naming convention and current boundaries of the World Bank and | |

of the validation of the new dataset, I recommend presenting information on the ratio of the sum of the components to the official values and interpreting any systematic discrepancies that are observed across regions, countries, or subdivisions. Second, I wonder whether the components the authors have constructed actually correspond to GDP components in all cases. GDP refers to value added, i.e., output value minus expenditure on intermediate inputs. I believe that some of the authors' components refer to output value (e.g., the crop and livestock estimates) whereas others refer to value added (e.g., the forestry estimates). If I am correct, then there is a conceptual inconsistency across the components that the authors must acknowledge and whose implications they must discuss

exclude disputed areas and a few small islands from the table. We may consider GAUL0 for boundaries and naming convention later, however these boundaries require a license.

We will also provide the **different components of priors in production values** in raster format in the data catalogue that we constructed which will allow users to better contextualize and ensure the fitness-for-use of their application(s). The maps are production values.

But we agree that the prior components will NOT correspond to real AgGDP components in all cases. We have added text in line 253 for the caveat: "**However, it should be noted that due to limitations in available data, we have some components in output values (crop, livestock, and fishery) whereas others in value added (forestry and hunting). This may result in discrepancies and inconsistencies.**"

We also have added text on WDI AgGDP value-added in line 225: "**The World Bank compiles these national accounts data following the International Standard Industrial Classification (ISIC) divisions 1-3 that includes agriculture, forestry and fishing. Given the challenges of compiling national accounts data across the world, limitations include the exclusion of unreported economic activity in the informal or secondary economy. In particular, agricultural output in developing countries may not be reported due to issues such as, natural losses, self-consumption or not exchanged for money. Despite best efforts, agricultural production may be estimated indirectly leading to approximations that are different than the true values. \footnote{See**

| | \href{https://data.worldbank.org/indicator/NV.AGR.TOTL.ZS}{World Bank WDI} for more details on metadata and limitations}" | |
|---|---|---|
| Line 241: The authors need to explain why they have chosen two drought indicators instead of one. Are two indicators necessary? If the purpose of the drought analysis is to illustrate the use and value of the new agricultural GDP dataset, then why not use only one? Moreover, given global concerns about climate change, why not illustrate use of the new dataset by using a forward-looking indicator of climate-change risks? The SPEI and WCI indicators are backward-looking, which makes them of dubious value given that climate change is altering drought risks. | We moved the illustration to a separate section and moved figures to the appendix for interested readers. The selection of backward-looking indicators is to match the temporal reference of the dataset using historical data. Indeed, climate-change risks are important with regards to agricultural GDP and future work can pursue these valuable links in time and space, especially as new input data become available. | We modified text |
| Table 2: I suspect that the correlations are not significantly different within some of the regions. I recommend adding information on the significance of the differences between the following pairs of correlations within each region: AgGDP/NTL vs. GDP/NTL, and AgGDP/NTL vs. POP/NTL. | Thank you. Table 2 showed country level correlations and we did explore the variation in differences as suggested. We have constructed a new **Table 2** highlighting the smaller correlations across AgGDP with NTL and POP at the **administrative 2 level** compared to the country level. Also, we have noted in line 300 that "**Sub-Saharan Africa and the Other Region show lower correlation values than other regions**." | We modified the text |
| Figure 7: Given that grid cells become smaller at higher latitudes, shouldn't the map show $/km2 instead of $? | We have taken into account that grids have different areas. We use the SPAM 5-arcminute grids to match with the data format of crop and livestock. | |
| Lines 306-307: The authors state, "One advantage of the cross-entropy is the volume preserving pycnophylactic property, which ensures the sum of the gridded data is the original value …." Spatial regression presumably violates this property. Does the analysis of predictive accuracy in the Brazil study by Thomas et al. (2019) indicate how much spatial regression violates it? In the current paper, the authors' comparison of the cross-entropy dataset to the naïve dataset based on rural population would be more compelling if | The spatial regression approach presented in Thomas et al. (2019) does produce a residual, however the model treats the residual value as a "fixed effect" for the aggregated unit as an unobserved characteristic of the unit. A post-estimation procedure adds the residual value evenly across the pixels within the aggregated unit to the predicted value at the pixel level to preserve the pycnophylactic property. | We modified the text |

| | | |
|---|---|---|
| Thomas et al. find that spatial regression violates the property a lot and thus is internally less consistent than the cross-entropy dataset. | | |
| Line 317: The authors state, "Since we cannot perform an evaluation of prediction accuracy for all countries …." Why not? I'm not saying they should perform such an evaluation. I am just unclear as to why they cannot perform it. Can they perform it for a subset of countries? | In the revised section **3.3 Validation**, we have modified text to clarify the data requirements for a global validation in **the first paragraph**. We did perform validation in the Brazil case and we have added discussion in detail in **the second paragraph in 3.3 Validation.** Additionally, we conducted a global comparison of our model and a rural population-based model in **the third paragraph in 3.3 Validation.** It shows that our model provides more information than the rural per capita model. | We modified the text |
| Lines 320-324:  Doesn't the finding that the naïve and cross-entropy maps are not significantly different imply that one might as well use the (presumably) simpler and more transparent naïve approach instead of the cross-entropy approach? I.e., what is the advantage of the cross-entropy approach over the naïve approach if the two approaches yield statistically indistinguishable results? Preservation of the pycnophylactic property? If so, can the authors provide information on the degree to which the naïve approach violates that property? | As discussed in the revised validation section, the correlation between predicted and true values of AgGDP at 5564 municipios was 0.91 for the cross-entropy model and 0.81 for the rural population model, representing a 12% improvement. Their MAD and RMSE also have obvious gaps. So, we think that the cross-entropy model is superior to the rural population model.

Even though estimates at the global level correlate well, AgGDP provides additional detailed information that is not derived from population estimates. The rural population model does not account for the components of AgGDP. Furthermore, estimates of rural p.c. AgGDP may not be reasonable in order to ensure the pycnophylactic property.

Also, Rural population model assumes proportional activity to population and require a standard definition of rural, which can pose challenges in global applications (Roberts et al., 2017). In addition, this dataset may attenuate modeling concerns of endogeneity when | We modified text |

| | using AgGDP along with population or night time lights. | |
|---|---|---|
| Line 334: The authors need to define "MAUP." | Modified text as **"the Modifiable Areal Unit Problem (MAUP) (Openshaw, 1981)"** in line 327. | |
| Lines 336-337: The authors write, "The data are most appropriate for applications at global, continental and regional scales (You and Wood, 2006)." Aren't the data also appropriate for applications in countries that contribute data from a relatively large number of subdivisions to the cross-entropy optimization (e.g., Thailand)? | Thank you. We have modified the text. The fitness for use is dependent on the application where the user should consider factors including: area of the grid cell of AgGDP, the number of subdivisions of AgGDP from the country, and uncertainty in the priors. It would work for countries that have many subnational data, e.g., Thailand which has data for 76 subdivisions.

In line 317-322:
**" However, decisions regarding the use of the data at smaller spatial extents should be made with caution and with consideration of the underlying assumptions and characteristics of the area in question. Users should take into account factors such as area of the grid cell of AgGDP, the number of subdivisions of AgGDP from the country, and assumptions in the priors (e.g. see shares of priors in Table B8). When input data contains multiple observations, the AgGDP dataset may still be suitable for use, as it is already standardized in grid cells, which may facilitate integration with other data."** | We modified the text |
| Line 381: Starting the "Conclusions" section with discussion of the drought analysis is odd given that the main contribution of the paper is the construction of the new gridded dataset. The current second paragraph in the section would work better as the starting paragraph. | Thank you. We have removed the paragraph on drought analysis. Now the conclusion is more focused on presenting the new dataset on AgGDP and only mentions the drought and water scarcity analysis as an illustrative example. | We modified the text |

| | | |
|---|---|---|
| The manuscript includes an Appendix B but no Appendix A. Is Appendix A missing, or is Appendix B mislabeled? | We used the .tex template where Appendix A is for figures and Appendix B is for tables. The revised manuscript has both figures and tables. | The updated text has both figures and tables |